# STRONGER-MAS: MULTI-AGENT REINFORCEMENT LEARNING FOR COLLABORATIVE LLMS

**Yujie Zhao**[1]  **Lanxiang Hu**[1]  **Yang Wang**[2]  **Minmin Hou**[2]
**Hao Zhang**[1]  **Ke Ding**[2]  **Jishen Zhao**[1]
[1]University of California, San Diego    [2]Intel Corporation

## ABSTRACT

Multi-Agent System (MAS) and Reinforcement Learning (RL) are both widely adopted to improve large language model (LLM) agentic performance. MAS strengthens task-specialized performance via role-based orchestration; RL leverages environment rewards to train stronger policies, such as Group Relative Policy Optimization (GRPO)-style optimization. Yet applying on-policy RL training to MAS is underexplored. While promising, it poses several challenges. On the algorithm side, Standard GRPO grouping assumptions fail in MAS because prompts differ by role and turn. On the system side, the training system needs to support MAS-workflow-based rollouts and on-policy updates for both single and multiple policy models. To address these issues, we introduce *AT-GRPO*, consisting of (i) an Agent- and Turn-wise grouped RL algorithm tailored for MAS and (ii) a system to support both single-policy and multi-policy training. Across game, plan, coding, and math tasks, AT-GRPO demonstrates substantial performance gains across diverse domains. Especially on long-horizon planning tasks, AT-GRPO boosts accuracy from a 14.0–47.0% single-agent RL baseline to 96.0–99.5%. Furthermore, it improves reasoning performance, with an average gain of 3.87–7.62% on coding and 9.0-17.93% on math.

**Code & Environment:**
https://github.com/pettingllms-ai/PettingLLMs

# 1 INTRODUCTION

Large Language Model (LLM) agents are task-specific workflows (Yao et al., 2023; Xi et al., 2023; Wang et al., 2023b) that utilize LLMs as key components for decision making (Shinn et al., 2023), action taking (Wang et al., 2023a), and tool use (Qian et al., 2025; Schick et al., 2023). LLM agents have demonstrated strong promises across various application domains, such as embodied control (Ahn et al., 2022; Wang et al., 2023a), software engineering (Tao et al., 2024; Yu et al., 2025), expert drug discovery (Liu et al., 2024; Inoue et al., 2024), and scientific ideation and hypothesis testing (Ghafarollahi and Buehler, 2024).

Today, two complementary approaches are widely used to improve the performance of LLM agents: multi-agent systems (MAS) and reinforcement learning (RL). RL treats the LLM as a policy and iteratively updates its weights to strengthen decision-making: at each iteration, the current model interacts with the environment, collects rule-based rewards, and then computes a policy optimization loss to update the parameters (Shao et al., 2024). In practice, this workflow requires a training stack that supports both scalable rollouts and online updates, e.g., VERL (Sheng et al., 2025) and AReaL (Fu et al., 2025). MAS typically employs prompt-only augmentation on a shared LLM policy for role-based coordination; practical deployments instantiate diverse workflows. Recent studies (Belcak et al., 2025; Chen et al., 2024; Wang et al., 2024) further highlight the potential benefits of role-specialized MAS, which adopts distinct models for different roles, enabling role-specialized policies in inference. However, the effectiveness of RL training on role-specialized MAS is underexplored.

A natural next step is to combine the two: using RL to train MAS, such that we gain both stronger learned policies, role-specialized collaboration. However, bringing RL into MAS raises two coupled

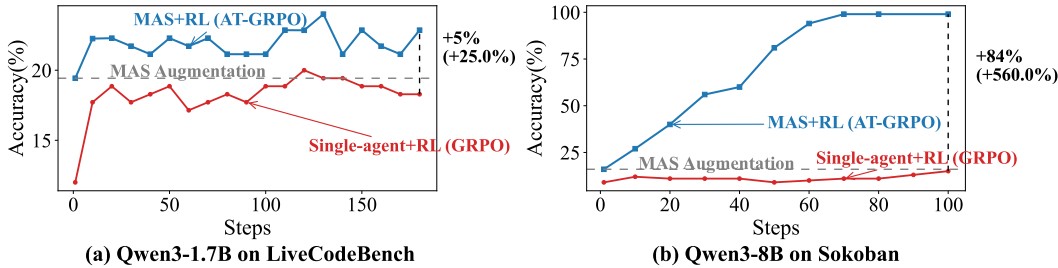

Figure 1: **MAS+AT-GRPO vs. Single-agent+GRPO.** The gray line denotes the prompt-only MAS baseline.

challenges. First, training a MAS may require concurrently launching multiple models, orchestrating inter-agent environment interactions, and performing independent on-policy parameter updates. But most existing on-policy RL frameworks for LLM agents only support a single model (Volcano Engine, 2025; Sheng et al., 2024; Fu et al., 2025). Second, rollouts from MAS are difficult to group. The advantage must be conditioned on interaction history and role to ensure fair credit assignment. Group-based RL objectives designed for a single agent (Volcano Engine, 2025; Qian et al., 2025; Feng et al., 2025) are not directly applicable to MAS.

To address these challenges, we first design *AT-GRPO*, an Agent- and Turn-wise grouped RL method that adapts group-relative optimization for MAS. Furthermore, we develop a novel training system to support on-policy RL for MAS. Our training system supports rollouts for diverse MAS workflows and enables on-policy RL training for both role-sharing policy and role-specific policies. We conduct extensive experiments on Qwen3 models across a range of representative agentic domains, including game, planning, coding, and mathematical reasoning. As highlighted in Fig. 1, AT-GRPO (blue) significantly outperforms single-agent GRPO (red). For instance, it achieves a 5.0% higher accuracy (+25.0% relative) on LiveCodeBench (with Qwen3-1.7B), while the improvement increases to 84.0% on Sokoban (with Qwen3-8B).

This paper makes the following key contributions:

- **AT-GRPO Algorithm.** We introduce an agent- and turn-wise grouped RL algorithm, AT-GRPO, and identify the substantial benefits of applying on-policy RL to MAS across diverse domains: planning, gaming, coding and mathematical reasoning tasks.
- **MAS Training System.** We design a novel training system to support (i) executing rollouts for diverse MAS workflows and (ii) performing on-policy RL updates for multiple policies.
- Our method delivers **consistent gains across diverse domains**. On long-horizon planning tasks, it overcomes a key bottleneck of single-agent RL, boosting accuracy from a 14–47% baseline to 96.0-99.5%. Furthermore, it also demonstrates gains on code and math benchmarks, with average improvements of 3.87–7.62% and 9.0–17.93%, respectively.
- Our **analysis** shows that (1) RL training on MAS reinforces role-specific specialization; (2) with MAS AT-GRPO, whether to choose a role-sharing policy or role-specialized policies needs to be determined by the task characteristics.

## 2 RELATED WORK

**Role-sharing vs. Role-specialized Policies in MAS.** A predominant approach in LLM-based MAS centers on a role-sharing architecture, where a single policy is shared across all agents. In these frameworks, such as AutoGen (Wu et al., 2023) and MetaGPT (Hong et al., 2024), role-specific behavior is elicited at inference time via prompt augmentation. More recently, research has begun to explore role-specialized policies. This shift is motivated by the observation that a single LLM's performance exhibits significant variance across domains (Chen et al., 2024; Wang et al., 2024; Belcak et al., 2025). Consequently, assigning distinct and more suitable models to specialized roles, as demonstrated by Ye et al. (2025); Belcak et al. (2025), has emerged as a promising direction for enhancing performance. Despite this architectural evolution, recent surveys (Cemri et al., 2025; Guo et al., 2024) indicate that most studies focus on inference-time design, leaving the potential of training MAS policies with RL largely underexplored.

**RL Training for MAS.** RL has become a key technique for LLMs agent training, using group-relative and rule-based rewards to enhance reasoning, long-horizon planning, game, and tool use

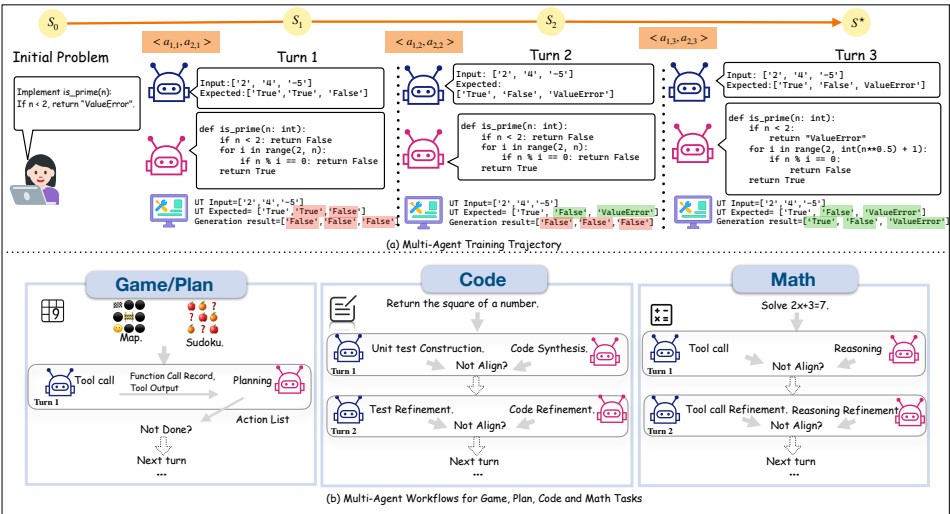

Figure 2: **MAS workflow across different domains.** (a) Role-based coordination: code generation via a coder–tester loop. (b) Different task-specific workflows for Game/Plan, Code, and Math; see Sec. 5.1 and Appendix C.2 for workflow details.

(Feng et al., 2025; Wang et al., 2025b; Qian et al., 2025; Hu et al., 2025). These approaches, however, predominantly operate within a single-agent framework. While a growing body of work attempts to extend RL to Multi-Agent Systems (MAS), most efforts remain confined to limited interaction settings or fixed role structures. For instance, CURE (Wang et al., 2025a) focuses on co-evolving a Coder and Unit-Tester using a role-sharing policy specifically for code generation. Similarly, SPIRAL (Liu et al., 2025) employs self-play in zero-sum games using a single LLM, while MHGPO (Chen et al., 2025a) targets retrieval-augmented generation. MAPoRL (Park et al., 2025a;b) and CoRY Ma et al. (2024) train LLMs within fixed, homogeneous-role debate workflows. More recent works also exhibit limitations: MARFT Liao et al. (2025) [1] restricts agents to single-turn sequential interactions, and MARTI Zhang et al. (2025) merely introduces basic single-agent RL algorithms (e.g., GRPO) to the MAS setting. Critically, the implemention for these works is concentrated on the single domain of math. We include a more comprehensive comparison of these related works in Appendix A. In contrast, our study offers a more comprehensive solution. We propose a general algorithm designed for multi-turn, multi-agent environments. Unlike prior works, we conduct a thorough analysis and evaluation of both shared and role-specific policies across diverse MAS workflows and varying domains.

## 3 PRELIMINARIES

**MAS Setting.** The $N$-agent LLM system is modeled as a Markov game $\mathcal{M} = (\mathcal{S}, \{\mathcal{A}_i\}_{i=1}^N, \mathcal{T}, \{r_i\}_{i=1}^N, T, H)$, where $\mathcal{S}$ is the state space; $\mathcal{A}_i$ is the action space of agent $i$; The transition function $\mathcal{T}$ induces intra-turn micro-transitions where $s_{t,0} = s_t$ and $s_{t,i} = \mathcal{T}(s_{t,i-1}, a_{t,i}, i)$, culminating in $s_{t+1} = s_{t,N}$. The reward for agent $i$ is given by $r_i : \mathcal{A}_i \to [0, 1]$, and the turn horizon $T$, the optimization step horizon $H$. At each turn $t$, agent $i$ receives an observation summarizing the environment state and interaction history $h_t$, $o_{t,i} = o_i(s_t, h_t)$. Each agent $i$ is implemented with a role-specific prompt template $\mathsf{P}_i(\cdot)$. Let $\Theta = \{\theta^{(m)}\}_{m=1}^M$ denote the set of LLM parameter vectors, with $1 \le M \le N$, and let $\sigma : \{1, \ldots, N\} \to \{1, \ldots, M\}$ assign each agent to an LLM. We treat one LLM rollout (a token sequence) as a single macro-action $a_{t,i}$. A *turn* is one full interaction in which all agents emit macro-actions to the environment. A *step* denotes one optimization update to the parameter set $\Theta$ during training.

**MAS Workflow.** Following prior work (Wang et al., 2025a; Ahn et al., 2022; Chen et al., 2025c), we employ domain-specific MAS workflows, as shown in Fig. 2. Our experiments confirm that this prompt-only method outperforms a single-agent baseline (see Tab. 1 and 2 in Sec. 5.2).

---

[1] We compare against MARFT v3, the latest preprint available prior to the completion of this work.

**Group-based RL.** Methods for LLM agentic training with group-relative advantages (Feng et al., 2025; Wang et al., 2025b; Qian et al., 2025) operate by first sampling $K$ candidate actions $\{a_t^{(c)}\}_{c=1}^K$ for a given prompt. Each action is evaluated to obtain a rule-based reward $R(a_t^{(c)})$, forming a comparison group: $G = \{(a_t^{(1)}, R(a_t^{(1)})), \ldots, (a_t^{(K)}, R(a_t^{(K)}))\}$. For each action $a_t^{(c)}$ in this group, the relative advantage is then defined as its mean-centered and normalized return.

$$A_g\big(a_t^{(c)}\big) = \frac{R(a_t^{(c)}) - \text{mean}\Big(\{R(a_t^{(c)})\}_{c=1}^K\Big)}{F_{\text{norm}}\Big(\{R(a_t^{(c)})\}_{c=1}^K\Big)}, \tag{1}$$

**Role-sharing vs. Role-specialized Policy Optimization.** We distinguish between two optimization regimes, role-sharing and role-specialized, both of which initialize policies from the same base model. During rollouts, each agent $i$ generates a dataset $\mathcal{D}_i$, which consists of sample groups. A single group $g$ is composed of a shared observation context $o_g$ and $K$ candidate actions with their corresponding advantages, denoted as $g = \{i, a_g^{(c)}, A_g^{(c)}\}_{c=1}^K$. The core difference between the two regimes lies in how the training data is batched. A minibatch $\mathcal{B}_m = \bigcup_{i : \sigma(i)=m} \mathcal{D}_i$. for a specific policy $\theta^{(m)}$ is constructed by pooling the datasets from all agents assigned to it:

$$\mathcal{L}(\theta^{(m)}) = -\mathbb{E}_{g \in \mathcal{B}_m}\left[\frac{1}{K}\sum_{c=1}^K \min\Big(r_g^{(c,m)}(\theta^{(m)})\, A_g^{(c)}, \text{clip}\big(r_g^{(c,m)}(\theta^{(m)}), 1-\varepsilon, 1+\varepsilon\big) A_g^{(c)}\Big)\right] \tag{2}$$

where $r(\theta) = \frac{\pi_\theta(o_i|q)}{\pi_{\theta_{old}}(o_i|q)}$. *Role-sharing policy (M=1):* All agents share a single policy $\theta^1$. The training batch is the union of data from all agents, $\mathcal{B}_1 = \bigcup_{i=1}^N \mathcal{D}_i$, and is used for a single joint update: $\theta^1 \leftarrow \theta^1 - \eta\nabla_{\theta^1}\mathcal{L}(\theta^1)$.

*Role-specialized policies (M = N):* Each agent $i$ has a distinct policy $\theta^{(i)}$, such that $\sigma(i) = i$. Each policy is updated independently on $\mathcal{B}_i = \mathcal{D}_i$, and update policy: $\theta^{(i)} \leftarrow \theta^{(i)} - \eta\nabla_{\theta^{(i)}}\mathcal{L}(\theta^{(i)})$.

## 4 METHOD

### 4.1 ALGORITHM DESIGN: AT-GRPO

GRPO's advantage calculation (Eq. 1) hinges on a fair comparison among all candidates within a group. This fairness is enforced by the reward mechanism itself. As illustrated in Fig. 2 (top), token-level scoring assigns credit to the generated response tokens (Reward Mask=1), while the prompt tokens receive no credit (Reward Mask=0). Since the advantage is determined solely by the quality of the response, a valid and fair comparison is only possible

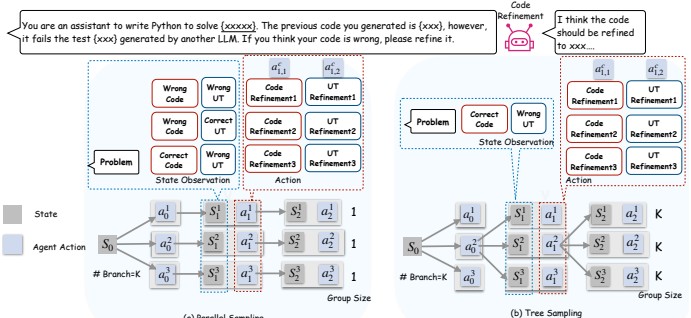

Figure 3: Two sampling schemes. **(a) In parallel sampling**, trajectories are sampled but incomparable, leading to groups of size 1. **(b) In tree sampling**, branching at each turn forms a valid comparison group of size $K$.

when all responses in a group originate from an identical prompt. Consequently, single-agent LLM-RL methods(Wang et al., 2025b; Qian et al., 2025; Feng et al., 2025) typically form groups by sampling multiple responses to the same question.

In MAS, however, a "prompt" is not only a question description, but also embeds the role-specific context and cross-agent interaction history. For example, in the code tasks depicted in Fig. 2 (a) , where the workflow entails a coder-tester loop: one agent synthesizing code and the other creating

---

**Algorithm 1** AT-GRPO: Agent- and Turn-wise MAS RL Training

---

**Require:** Markov game $\mathcal{M}$, policies $\Theta = \{\theta^{(m)}\}_{m=1}^M$, role mapping $\sigma$, sampling temperature $T_{\text{samp}}$, branches $K$, total steps $S$, batch size $E$, turn horizon $T$, termination condition $\mathcal{I}_{\text{term}}$.
  /*– Termination helper: returns true if horizon reached or env signals done –*/
 1: **for** training step $s = 1, \ldots, S$ **do**
  /*– Phase 1: On-Policy Rollout & Data Collection –*/
 2:  Initialize per-agent datasets $\{\mathcal{D}_i\}_{i=1}^N \leftarrow \emptyset$. Resample $E$ environments.
 3:  **for each** environment instance $e \in \{1, \ldots, E\}$ **in parallel do**
 4:   **for** $t = 0$ **to** $T - 1$ **do**
 5:    $s_{t,0,e} \leftarrow s_{t,e}$             ▷ Initialize micro-step state
 6:    **for each** agent $i \in \{1, \ldots, N\}$ **do**
 7:     $\forall c \in \{1, \ldots, K\}, a_{t,i,e}^{(c)} \sim \pi_{\theta^{(\sigma(i))}}(\cdot \mid o_{t,i,e}; T_{\text{samp}})$; compute $r_{t,i,e}^{(c)}$ (Eq. 3)
 8:     Define group key $g \leftarrow \text{hash}(e, i, t)$ and compute advantages $\{A_g^{(c)}\}_{c=1}^K$ (Eq. 1).
 9:     Append $(g, o_{t,i,e}, \{a_{t,i,e}^{(c)}\}_{c=1}^K, \{A_g^{(c)}\}_{c=1}^K)$ to $\mathcal{D}_i$.
10:     $c^\star \leftarrow \arg\max_c r_{t,i,e}^{(c)}; \quad a_{t,i,e} \leftarrow a_{t,i,e}^{(c^\star)}$. *(Tree-structured sampling.)*
11:     $s_{t,i,e} \leftarrow \mathcal{T}(s_{t,i-1,e}, a_{t,i,e}, i)$    ▷ Agent-wise micro-transition
12:    **end for**
13:    $s_{t+1,e} \leftarrow s_{t,N,e}$           ▷ End-of-turn state
14:    **if** $\mathcal{I}_{\text{term}}(s_{t+1,e})$ **then break**
15:    **end if**
16:   **end for**
17:  **end for**
  /*– Phase 2: Per-Model Policy Update –*/
18:  **for each** model $m \in \{1, \ldots, M\}$ **in parallel do**
19:   Construct per-model batch $\mathcal{B}_m$, loss $\mathcal{L}(\theta^{(m)})$ on $\mathcal{B}_m$ using Eq. 2 and update policy $m$.
20:  **end for**
21: **end for**

---

unit tests, and they iteratively refine the output until alignment (Fig. 3, top), the turn-2 refinement prompt already contains the turn-1 code, unit tests, and role-specific prompt format, so prompts differ across turns and roles. We therefore adopt agent-wise and turn-wise grouping as a natural extension of tabular-wise group-normalized advantages in GiGPO (Feng et al., 2025) to the multi-agent setting: candidates share the same role and turn position, ensuring prompt identity for valid GRPO advantage comparisons.

However, agent- and turn-wise grouping introduces a new question. If we follow the common parallel sampling used by prior agentic RL—sample $K$ full trajectories from the initial state/problem (Fig. 3 (a), bottom), each group size = 1 when $t > 1$: no other sample shares the identical prompt. GRPO therefore eliminates its variance-reduction effect and yields unstable updates. To address these challenges, we develop AT-GRPO (see Alg. 1) with three key ideas: *tree-structured sampling*, *agent– and turn-wise grouping*, and *agent-wise credit assignment*.

**Tree-structured Sampling.** At each turn $t$, for each agent $i$, we sample $K$ candidate actions and their corresponding rewards from the current state (Alg. 1, line 7). The advantages for these $K$ candidates are then calculated within this group (line 9). Subsequently, the full data tuple—containing the group key, observation, $K$ actions, and their $K$ advantages—is added to a dataset $D_i$ specific to the policy of the acting agent $i$ (line 10). To proceed with the environment rollout, we greedily select the candidate with the highest reward to be the executed action (line 11). This greedy selection strategy concentrates exploration on coordination-critical decisions and helps maintain a balanced mix of positive and negative samples, which stabilizes the learning optimization.

**Agent– and Turn-wise Grouping.** We group experiences based on the acting agent and the turn number within each parallel environment instance. Operationally, we implement this by defining a unique group key $g$ for each agent $i$ at each turn $t$ in each environment $e$ using a lightweight hash function (Alg. 1, line 8). All data generated from the $K$-branch sampling at that step, including the observation and the calculated advantages, is stored together under this group key (line 10). During

the policy update phase, these collected data groups are used to construct per-model training batches for the final optimization step (lines 20–21).

**Agent-wise Credit Assignment.** Inspired by mixed-reward designs in cooperative Multi-Agent RL (Mao et al., 2020; Sheikh and Bölöni, 2020), we assign credit using a mixture of global and local rewards. At each turn $t$, the environment provides a global team reward $r^{\text{team}}$ and an agent-specific local reward $r_i^{\text{loc}}$ that evaluates its subtask performance. These components are combined using a hyperparameter $\alpha$ to form the final reward for agent $i$:

$$r_{t,i} \;=\; \alpha\, r_t^{\text{team}} \;+\; r_{t,i}^{\text{loc}} \tag{3}$$

This formulation balances a shared team objective with role-specific incentives. For instance, in a coder-tester MAS, the team reward $r^{\text{team}}$ is the pass rate of the generated program on a set of golden unit tests. The local rewards $r_i^{\text{loc}}$ are tailored to each role: the coder is rewarded for its own code's pass rate, while the tester is rewarded based on the pass rate of a golden reference implementation against its generated tests. Detailed reward designs for all tasks are provided in Appendix C.1.

### 4.2 MAS Training System

Mainstream RL post-training frameworks for LLMs, e.g., TRL (von Werra et al., 2020), VERL (Sheng et al., 2024), AReaL (Fu et al., 2025), and OpenRLHF (Hu et al., 2024) primarily support single-agent RL training, which typically involves: a single agent-environment interaction pattern, a policy operating, and a single LLM resource pool. This makes it difficult to (i) train multiple models in on-policy RL, (ii) maintain clean on-policy training data, and (iii) support diverse MAS workflow.

We introduce a novel MAS training system to overcome these challenges and enable AT-GRPO . By allocating an independent resource pool to each model, our system is designed to support the concurrent on-policy training of multiple policies. The system, depicted in Fig. 4, consists of the following components:

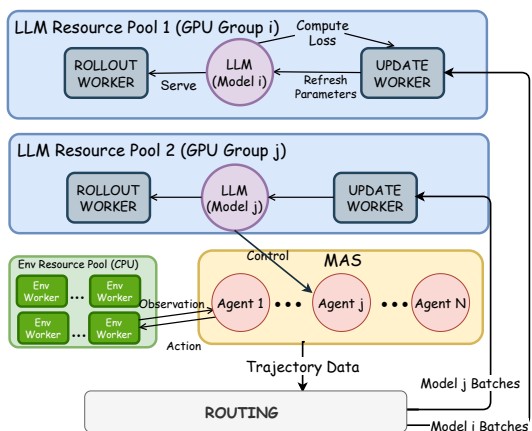

Figure 4: **MAS training system.** Each LLM $m$ has a GPU-pinned model pool with a Rollout-Worker and an UpdateWorker. A CPU environment pool hosts envworkers that execute environment steps. Trajectories are routed to the corresponding UpdateWorker.

**LLM Resource Pools (GPU).** Each policy is managed within an independent resource pool. Following HybridFlow-style (Sheng et al., 2025), each pool comprises two workers: a *RolloutWorker* for inference and an *UpdateWorker* for optimization. During the rollout phase, all policies interact collectively according to the Alg. 1 and MAS workflow; Once collected, each trajectory is routed to the corresponding UpdateWorker, maintaining an on-policy learning regime for every policy.

**Environment Execution (CPU) and Data Flow.** Environment steps run in a fleet of CPU *EnvWorkers*, each managing a single sandboxed instance to ensure safety and reproducibility (seeding, wall-clock timeouts, IO quotas, and deterministic tool harnesses). This one-actor-per-instance mapping efficiently supports thousands of concurrent rollouts in parallel. EnvWorkers stream observations, tool logs, and rule-based rewards to a *Router*. The Router dispatches collected experience based on policy assignment: experiences generated by an agent $i$ are sent to the Updateworker of its designated policy $\sigma(i)$.

## 5 Experiments

### 5.1 Datasets and Models.

1. **Experimental Setup.** We train and evaluate Qwen3 models at 1.7B and 8B in the no-thinking mode (Yang and the Qwen Team, 2025). All runs use a single node with $8\times$ H100 GPUs. The rollout

sample size is $K=4$ and the turn horizon is $T=4$ for both multi-agent (MA) and single-agent (SA) settings. The reward-mixing coefficient is $\alpha=1$ without further tuning. Full training details appear in Appendix C.1.

**2. Baselines.** We evaluate five variants (all initialized from the same base model): (a) Single Agent (prompt-only): one frozen LLM solves the task end-to-end; (b) Single Agent + GRPO: as (a) but trained with GRPO (Shao et al., 2024); (c) MAS (prompt-only): role-specialized prompting over a frozen, role-sharing backbone; (d) MAS + RL (role-sharing policy): all roles share one policy and pooled trajectories update it jointly; (e) MAS + RL (role-specialized policies): samples are routed by role and each policy is optimized independently (no parameter sharing).

**3. Task Setups and Baselines.** To ensure a fair comparison, we align all environmental observations and reward signals across both MA and SA settings. While both paradigms utilize the same role-specific reward functions, the sole distinction is that the MA framework involves multiple agents capable of discussion. Detailed prompt templates and reward specifications are provided in Appendix B and C.2.

**Single Agent Variants.** We also evaluated a multi-turn SA variant for both Code and Math (see Appendix F), where a single agent repeatedly revises its own output until self-consistency. This setup is inherently less natural: the agent receives no additional environmental signal or cross-agent feedback, and the interaction pattern deviates from the QA-style pretraining regime of LLMs. Reflecting this mismatch, the multi-turn SA variant brought no empirical improvement and sometimes slightly degraded performance relative to the standard single-turn SA baseline.

**Code.** The environment observation is restricted to the problem description. The MA setting employs a dual-role debating mechanism: a *Tester* generates unit tests and a *Coder* generates the code. They refine their outputs iteratively until alignment is reached or the maximum turn limit is met. The natural SA baseline employs a *Coder* to generate the solution directly, as no other environmental feedback and other agent's output is available.

**Math.** The environment observation consists is restricted to the problem. MA uses Dual-role debating MAS: a *Tool-User* (utilizing code interpreters) and a *Reasoner* (performing direct reasoning) until alignment or the maximum turn limit is met. The SA setting uses a *Reasoner* to derive the answer directly, as no other environmental feedback and other agents' response.

**Planning and Game.** The environment observations are game states of the current turn. The MA setting employs a collaboration mechanism: a *Tool-User* (executing the tools) and an *Executor* (verifying tool outputs and executing actions). The SA employs an *Executor* to perform actions. Both settings share identical termination conditions based on goal satisfaction or the turn budget.

**4. Training and Evaluation Datasets.** **Sudoku and Sokoban.** We evaluate our method on gaming tasks: a $4\times4$ Sudoku and a $6\times6$ Sokoban. We use instances with an automatic checker, following the symbolic task setup of SYMBENCH (Chen et al., 2025c). To ensure a fair evaluation, we generate distinct training and validation sets using different random seeds and verify there is no overlap.

**Plan-Path.** We use a $10\times10$ grid-based Plan-Path environment. This follows the checker-backed symbolic task setup in CodeSteer's SymBench (Chen et al., 2025c). To separate training and validation, we generate the two splits with distinct random seeds and verify no duplication.

**Code Generation.** For training, we adopt size-specific corpora: the 1.7B Qwen model is trained on the APPS training split (introductory-difficulty subset) (Hendrycks et al., 2021), while the 8B model is trained on CodeContests (DeepMind, 2024). For model-generated code, we use the dataset's golden unit tests to score correctness; for model-generated UT, we use the dataset's golden reference solutions to compute the reward. For evaluation, we use three widely adopted coding benchmarks spanning interview-style and contest-style settings: APPS (Hendrycks et al., 2021), LiveCodeBench-v6 (White et al., 2024), and CodeContests (DeepMind, 2024).

**Mathematical Reasoning.** We train on the Polaris-Dataset-53K (An et al., 2025) and evaluate on several standard mathematical reasoning benchmarks. For validation, we use AIME24/AIME25 (Mathematical Association of America & AoPS Community, 2024; 2025) and OLYMPIADBENCH (He et al., 2024). All math tasks use verifier-checked numeric scoring.

Table 1: **Qwen3 1.7B** results on game, planning, coding, and math.

| Method | Game | | Plan | Code | | | Math | | |
| --- | --- | --- | --- | --- | --- | --- | --- | --- | --- |
| | Sudoku | Sokoban | Plan-Path | LiveCodeBench | APPS | CodeContests | AIME24 | AIME25 | OlympiadBench |
| Single agent | 7.00 (+0.00) | 0.00 (+0.00) | 5.00 (+0.00) | 11.60 (+0.00) | 16.20 (+0.00) | 3.60 (+0.00) | 13.40 (+0.00) | 9.80 (+0.00) | 22.20 (+0.00) |
| Single agent + GRPO | 29.00 (+22.00) | 3.00 (+3.00) | 11.00 (+6.00) | 18.80 (+7.20) | 17.00 (+0.80) | 3.00 (-0.60) | 10.00 (-3.40) | 6.70 (-3.10) | 23.80 (+1.60) |
| MAS | 69.00 (+62.00) | 0.00 (+0.00) | 10.00 (+5.00) | 19.00 (+7.40) | 16.60 (+0.40) | 3.60 (+0.00) | 13.30 (+-0.10) | 13.00 (+3.20) | 35.90 (+13.70) |
| MAS + GRPO | 87.00 (+80.00) | 1.00 (+1.00) | 82.00 (+77.00) | 20.60 (+9.00) | 17.60 (+1.40) | 4.80 (+1.20) | 13.30 (+-0.10) | 16.70 (+6.90) | 35.00 (+12.80) |
| MAS + AT-GRPO w/ shared policy | **99.00** (+92.00) | 10.00 (+10.00) | 96.00 (+91.00) | 20.90 (+9.30) | 17.60 (+1.40) | 4.80 (+1.20) | **16.70** (+3.30) | 16.70 (+6.90) | **39.60** (+16.80) |
| MAS + AT-GRPO w/ per-role policies | **99.00** (+92.00) | **11.50** (+11.50) | **97.00** (+92.00) | **24.00** (+12.40) | **18.60** (+2.40) | **7.80** (+4.20) | 13.30 (+-0.10) | **18.30** (+8.50) | 35.20 (+13.00) |

Table 2: **Qwen3 8B** results on game, planning, coding, and math.

| Method | Game | | Plan | Code | | | Math | | |
| --- | --- | --- | --- | --- | --- | --- | --- | --- | --- |
| | Sudoku | Sokoban | Plan-Path | LiveCodeBench | APPS | CodeContests | AIME24 | AIME25 | OlympiadBench |
| Single agent | 48.00 (+0.00) | 9.00 (+0.00) | 12.00 (+0.00) | 22.80 (+0.00) | 30.20 (+0.00) | 15.75 (+0.00) | 18.30 (+0.00) | 20.00 (+0.00) | 55.00 (+0.00) |
| Single agent + GRPO | 54.00 (+6.00) | 14.00 (+5.00) | 47.00 (+35.00) | 25.70 (+2.90) | 37.00 (+6.80) | 12.12 (-3.63) | 18.30 (+0.00) | 26.67 (+6.67) | 54.80 (-0.20) |
| MAS | 72.00 (+24.00) | 16.00 (+7.00) | 71.00 (+59.00) | 28.00 (+5.20) | 44.40 (+14.20) | 17.60 (+1.85) | 36.60 (+18.30) | 30.00 (+10.00) | 56.50 (+1.50) |
| MAS + GRPO | 99.00 (+51.00) | 30.00 (+21.00) | 96.00 (+84.00) | 24.20 (+1.40) | 40.20 (+10.00) | 10.30 (-5.45) | 33.30 (+15.00) | 26.67 (+6.67) | 53.20 (-1.80) |
| MAS + AT-GRPO w/ shared policy | **99.50** (+51.50) | 96.00 (+87.00) | 93.00 (+81.00) | 30.28 (+7.48) | 45.80 (+15.60) | **18.10** (+2.35) | 50.00 (+31.70) | 35.20 (+15.00) | **56.80** (+1.80) |
| MAS + AT-GRPO w/ per-role policies | 99.00 (+51.00) | **98.00** (+89.00) | **96.00** (+84.00) | **33.10** (+10.30) | **46.50** (+16.30) | **18.10** (+2.35) | **57.00** (+38.70) | **40.00** (+20.00) | 56.60 (+1.60) |

Parentheses denote gain over the Single Agent baseline; best and second-best results per column are highlighted.

## 5.2 RESULTS AND ANALYSIS

We evaluate AT-GRPO across four distinct domains (game, planning, code, and math) using two model scales (Qwen3 1.7B and 8B). To contextualize its performance, we benchmark against all the variants described in Sec. 5.1. Tab. 1 and Tab. 2 summarize our main results.

**MAS + AT-GRPO consistently yields substantial performance gains, especially in long-horizon planning tasks.** MAS + AT-GRPO elevates the success rate from a 14–47% range for the single-agent baseline to 96.0–99.5%. By analyzing the dialogue records between agents, we find this dramatic improvement stems from an emergent collaboration: the tool agent learns to generate correct algorithms (e.g., BFS, $A^\star$ search), while the plan agent provides crucial oversight, interpreting execution outcomes and delivering the corrective final action list. On-policy RL training within the MAS enhances inter-agent coordination. Conversely, training agents in isolation results in only limited improvement, as detailed in our ablation study (Sec. 5.4, Tab. 4). Furthermore, on the coding and math benchmarks, our approach yields consistent gains, with absolute gains over the baseline ranging from +2.35 (CodeContests) to +16.30 (APPS) in coding, and from +1.80 (OlympiadBench) to +38.70 (AIME24) in math. We hypothesize two reasons: (1) Base models like Qwen3 have already been extensively trained for these common domains, as noted in their official reports (Yang and the Qwen Team, 2025), potentially leading to performance saturation. (2) The diverse nature of problems within these domains presents a greater challenge for improvement via RL training.

**With MAS AT-GRPO, whether choosing a role-sharing policy or role-specialized policies should be determined by the task characteristics.** Role-specialized policies involve a fundamental trade-off: training each agent exclusively on its own data fosters deep specialization, but prevents access to potentially useful data from other roles. Our findings indicate that the optimal resolution to this trade-off depends on the task characteristics. We observe clear benefits for role specialization in the coding domain, where the Coder and Tester functions are highly distinct. This separation allows each agent to hone its specific skills, improving the average accuracy by 3.05 points with the Qwen3 1.7B.In contrast, the roles in the math domain exhibit greater functional overlap, meaning a shared policy can sometimes be superior. For instance, with the Qwen3 1.7B model on OlympiadBench, the shared policy achieves a 39.60% accuracy, surpassing the 35.20% from per-role policies. This suggests the Tool agent, which must often perform reasoning to execute tool

calls, benefits from the Reasoner's training data. For game/plan tasks, this choice becomes moot, as both configurations already achieve near-optimal, saturated performance (e.g., 99.50 on Sudoku).

**Limitations of MAS-GRPO.** Empirical results in Tab. 1 and 2 indicate that directly applying GRPO to MAS often results in performance degradation. Notably, Qwen3-8B exhibits suboptimal results CodeContests ($17.60 \rightarrow 10.30$) and OlympiadBench ($56.50 \rightarrow 53.20$). We attribute this to the violation of the identical-state assumption: as multi-turn interaction histories diverge, the group-averaged baseline incorrectly aggregates heterogeneous states. This structural misalignment biases advantage estimation and destabilizes optimization.

**Scalability Analysis with Collaborative Agents.** To investigate scalability, we deploy a modular MAS architecture comprising Reasoners, Tool-Users, and a Judge (Fig. 5(a)). By varying the number of concurrent Reasoners ($N$) and Tool-Users ($M$), we scale the total agent count ($M + N + 1$) to expand the exploration space. Fig. 5(b) demonstrates

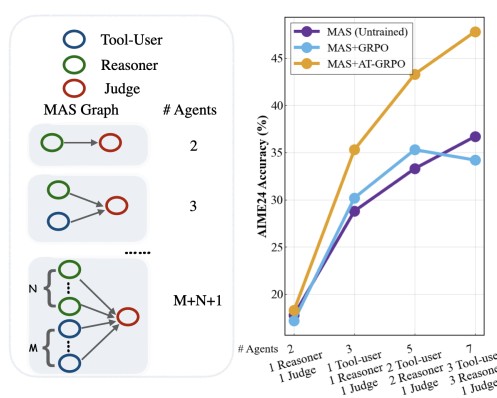

(a) Scalable MAS Interaction Graphs. (b) Accuracy Scaling w.r.t. Agent Count.

Figure 5: (a) The system aggregates outputs from an ensemble of $N$ Reasoners and $M$ Tool-Users into a Judge. The total agent count scales as $M + N + 1$, allowing for flexible resource allocation. (b) Evaluation on AIME24 (using Qwen3-8B).

the scalability of MAS+AT-GRPO. While the baseline MAS+GRPO fails to scale effectively—saturating at 34.1% accuracy in the 7-agent regime—our method successfully leverages the increased ensemble size, achieving a continuous performance gain from 18.2% to 47.7%. This confirms that MAS+AT-GRPO can effectively scale across multiple agents without hitting the coordination bottlenecks observed in baselines. For a detailed analysis of computational efficiency and complexity, please refer to Appendix G.

## 5.3 COMPARISON WITH OTHER MARL FRAMEWORKS

To assess the efficacy of our framework, we conducted ablation studies against representative baselines: As summarized in Tab. 3. we benchmark against three representative baselines: MAPORL (Park et al., 2025b), MARFT (Liao et al., 2025), and CURE Wang et al. (2025a). For fair comparison, we utilize identical base models and dataset splits. Our analysis highlights the advantages of two distinct MAS features: **heterogeneous agent roles** and **multi-turn iterative interaction**, as summarized in Tab. 3.

**Comparison with MAPORL.** We compare our approach with MAPORL using the Phi-3-mini-128k (3.4B) on gsm8k

Table 3: Comparison with existing MARL frameworks. We report Accuracy (%) for math/logic tasks and Pass@1 (%) for code tasks.

| Math: Accuracy (%) | | | |
|---|---|---|---|
| **Backbone** | **Method** | **Config** | **Acc.** |
| Phi-3-mini | Vanilla Baseline | Zero-shot | 65.0* |
| | MAPORL | Trained | 81.0* |
| | **Ours (MAS)** | **Untrained** | **84.4** |
| | **Ours (MAS+AT-GRPO)** | **Trained** | **88.7** |
| Qwen2.5-Coder-3B-Instruct | Vanilla Baseline | Zero-shot | 76.8* |
| | MARFT | Trained | 78.7* |
| | **Ours (MAS)** | **Untrained** | **84.4** |
| | **Ours (MAS+AT-GRPO)** | **Trained** | **87.1** |
| Code: Pass@1 (%) | | | |
| **Backbone** | **Method** | **CodeContests** | **LiveCodeBench** |
| Qwen-2.5-7B-Instruct | Vanilla Baseline | 22.8* | 26.9* |
| | CURE | 25.9* | 31.2* |
| | **Ours (MAS)** | **30.3** | **30.4** |
| | **Ours (+AT-GRPO)** | **34.2** | **35.3** |

\* Results cited from original papers.

dataset Cobbe et al. (2021). While MAPORL relies on a debating mechanism among homogeneous agents, our framework realizes role heterogeneity—synergizing a *Reasoning Agent* with a *Tool-use Agent* for verification. This structural specialization proves superior: our untrained MAS achieves 84.4%, outperforming the trained MAPORL (81.0%). With AT-GRPO training, our performance further improves to 88.7%.

**Comparison with MARFT and CURE.** This comparison highlights the critical efficacy of iterative alignment over single-turn workflows. In math reasoning ( Qwen2.5-Coder-3B-Instruct), while MARFT relies on single-turn preference optimization, our framework leverages multi-turn interactions to facilitate active error correction and ambiguity resolution. Consequently, our inference-only MAS (84.4%) significantly outperforms the trained MARFT (78.7%), confirming that an extended

reasoning horizon contributes more to robustness than single-step alignment; AT-GRPO training further amplifies this to 87.1%. A similar structural advantage is evident against CURE in code generation (Tab. 3). While CURE generates code and unit tests in a single turn, without utilizing them for self-correction. Our framework establishes a self-refinement cycle. This enables iterative debugging using generated tests, boosting CodeContests accuracy from 22.8% (vanilla) to 30.3% (untrained), surpassing the CURE baseline (25.9%), and ultimately reaching 34.2% with training.

## 5.4 ABLATION STUDY

To further investigate the contributions of our core training components, We also conducted an ablation study with results summarized in Tab. 4 and Fig. 6. Our analysis yields several observations.

First, **on-policy RL training within a MAS environment is critical for effective collaboration**. As shown in Tab. 4, training agents in a single-agent (SA) setting offers limited benefits: while individual agents improve their specialized skills (achieving 11.00 and 14.50 accuracy, respectively), their performance when combined in a MAS is only marginally better, reaching just 16.00. In stark

Table 4: Plan-Path (Qwen3-1.7B) ablation. Performance gain $\Delta$ over the single agent baseline.

| Method | Acc.(%) | $\Delta$ |
|---|---|---|
| Single agent | 5.00 | – |
| Training tool agent in SA, eval in SA | 11.00 | +6.00 |
| Training code agent in SA, eval in SA | 14.50 | +9.50 |
| Training in SA, eval in MAS | 16.00 | +11.00 |
| **MAS RL (role specific policies), eval in MAS** | **96.00** | **+91.00** |
| *w/ Swapped Policies* | 6.00 | +1.00 |

contrast, training the agents jointly within the MAS environment boosts accuracy to 96.00. This vast performance gap demonstrates that multi-agent training is essential. It not only allows agents to co-evolve highly specialized abilities but also fosters the crucial inter-agent alignment and collaboration required for success.

Second, **RL training on MAS reinforces role-specific specialization.** We observe this across multiple metrics. As shown in Fig. 6 (a) for Qwen3 1.7B on Plan-Path, the learning rewards of both the planning and tool-using agents increase throughout training, suggesting coordinated co-evolution as each adapts to the other's improving policy. Consistent with the ablation, after training two role-specialized policies with our full method, swapping them induces a catastrophic drop from 96.0% to 6.0%, confirming that the agents have learned distinct and complementary functions that are not interchangeable. In our coding (LiveCodeBench) and math (AIME25) workflows, MAS interaction terminates when the two agents align (e.g.,

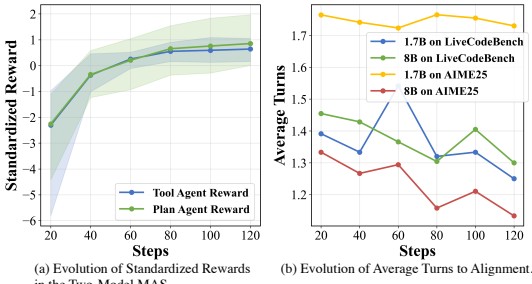

(a) Evolution of Standardized Rewards in the Two-Model MAS.

(b) Evolution of Average Turns to Alignment.

Figure 6: (a) Evolution of standardized rewards for the Tool and Plan agents in the role-specific MAS on Plan-Path with Qwen3 1.7B. Shaded bands show variability across runs. (b) Evolution of the average turns required to solve tasks.

tests pass or the reasoner and tool outputs agree). Accordingly, Fig. 6 (b) shows that the average number of turns needed to solve a task decreases over training, providing direct evidence that the agents achieve tighter alignment and collaborate more efficiently.

## 6 CONCLUSION

In this paper, we proposed **AT-GRPO**, an agent- and turn-wise grouped reinforcement learning algorithm tailored for on-policy training in MAS. To support this, we introduced a novel training system capable of managing diverse MAS workflows and performing on-policy updates for multiple policies. Our extensive experiments demonstrate that our method delivers consistent gains across diverse domains. On planning tasks, it overcomes a key bottleneck of single-agent RL by boosting accuracy from a 14–47% baseline to 96.0–99.5%. Furthermore, it improves reasoning performance with average gains of 3.87–7.62% on coding and 9.0–17.93% on math tasks. Our analysis reveals that RL training in MAS context reinforces role-specific specialization, with the choice between a shared or specialized policy contingent on the task's characteristics.

## 7 ETHICS STATEMENT

We study multi-agent reinforcement learning for large language models on planning, coding, and math tasks. Our experiments are purely computational and use public benchmarks (e.g., programmatically generated Plan-Path/Sudoku instances and widely available coding/math datasets) together with self-constructed simulators and verifiers. No human subjects, sensitive personal data, or proprietary content are involved. Code execution is performed in a sandboxed environment with restricted file I/O and no network access; tool calls are limited to deterministic checkers to prevent unintended side effects. While our methods are intended to improve reliability and sample-efficiency of agentic LLMs, we recognize dual-use risks common to autonomous systems (e.g., unsafe tool use or over-delegation). To mitigate these risks, we avoid external system operations, log all actions for auditability, and refrain from releasing any configurations that grant networked or privileged execution. We also note that base LLMs may encode societal biases that our training does not remove; results should therefore not be used for high-stakes decisions. We will release prompts, generators, and evaluation scripts to support reproducibility, subject to dataset licenses and safe-use guidelines.

## 8 REPRODUCIBILITY STATEMENT

To ensure the reproducibility of our results, we have made our datasets, code, and experimental details available. All datasets used in this study are publicly available; we provide detailed descriptions of these datasets and all data preprocessing steps in Sec. 5.2 and Appendix C.1. The source code used for our experiments is included in the supplementary material. Upon acceptance, we will release the complete, documented source code under a permissive open-source license to facilitate the reproduction of all presented results. Key hyperparameters, model architectures, and training configurations are also detailed in Appendix C.1.

## 9 USE OF LLM

During the preparation of this manuscript, a large language model was utilized to aid in polishing the grammar and improving the clarity of the text. The authors reviewed and edited all outputs to ensure the final content accurately reflects our original ideas and are fully responsible for all statements and conclusions presented.

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

# A EXTENDED COMPARISON OF RL TRAINING FOR LLM-BASED MAS

To complement the discussion on RL training for LLM-based multi-agent systems in Sec. 2, we provide an extended, axis-by-axis comparison of representative frameworks in Table 5. We contrast MAPoRL, MARFT, MARTI, CURE, and our method along several key design dimensions: (i) whether agents share a single policy or use role-specific policies; (ii) whether the interaction pattern is sequential, parallel, or a hybrid of both; (iii) whether the framework supports multi-turn interaction; (iv) whether agent roles are heterogeneous; (v) the number of evaluation task domains; and (vi) whether the underlying RL algorithm is designed as a general-purpose MAS training scheme rather than being tightly coupled to a single task.

Table 5: Comparison of RL-based LLM multi-agent training frameworks.

| Method | Policy sharing | Execution pattern | Multi-turn | Role heterogeneity | $\geq 2$ domains verification | Generic MAS RL algo. |
|---|---|---|---|---|---|---|
| MAPoRL Park et al. (2025a) | R | P | ✓ | ✗ | ✗ | ✗ |
| MARFT Liao et al. (2025) | R | S | ✗ | ✓ | ✗ | ✓ |
| MARTI Zhang et al. (2025) | R | S+P | ✓ | ✓ | ✗ | ✗ |
| CURE Wang et al. (2025a) | S | P | ✗ | ✓ | ✗ | ✗ |
| **Ours (StrongerMAS)** | S+R | S+P | ✓ | ✓ | ✓ | ✓ |

*Note.* For MARFT (Liao et al., 2025), we report the characteristics of version v3 of the framework, corresponding to the preprint available prior to the completion of this work.

As summarized in Tab. 5, StrongerMAS is the only RL-based LLM multi-agent training framework that simultaneously supports both shared and role-specific policies, hybrid sequential–parallel execution, multi-turn interaction, and heterogeneous agent roles, while being validated on *multiple* task domains and implemented as a general-purpose MAS RL algorithm. In contrast, MAPoRL (Park et al., 2025a) and CURE (Wang et al., 2025a) restrict policies to a purely role-shared setting and are tailored to specific tasks, MARFT Liao et al. (2025) restricts agents to single-turn sequential interactions, and MARTI Zhang et al. (2025) merely introduces basic single-agent RL algorithms (e.g., GRPO) to the MAS setting. . This combination of flexible workflow expressivity (S+R, S+P, heterogeneous roles) and broad, cross-domain evaluation makes StrongerMAS a more faithful and scalable abstraction for training cooperative LLM-based MAS.

# B REWARD DESIGN

## B.1 MATH REWARD DESIGN

We consider math QA with horizon $T$ and optional tool calls. Let $h_t$ be the dialogue/tool history at turn $t$. We adopt MATH-VERIFIER[2] as the checker front-end.

Define a numeric comparator with tolerance $\delta$:

$$\text{NUMEQ}_\delta(a,b) = \mathbf{1}\left\{|a-b| \leq \delta \ \text{ or } \ \frac{|a-b|}{\max(1,|b|)} \leq \delta\right\}, \quad \delta = 10^{-6}.$$

**Team reward.** Sparse pass at termination via numerical equality. We broadcast the same scalar reward to all turns:

$$r_t^{\text{team}} = \mathbf{1}\{\text{CHECKFINAL}_{\text{MATHVERIFIER+NUMEQ}}(h) = \text{pass}\} \in \{0,1\}, \quad \forall t.$$

**Local rewards.** Each agent $i$ at turn $t$ uses a masked convex combination of verifiable sub-scores $s_{\ell,t}^i \in [0,1]$:

$$r_{t,i}^{\text{loc}} = m_{t,i} \sum_{\ell \in \{\text{fmt,tool,step}\}} c_\ell^i \, s_{\ell,t}^i, \qquad \sum_\ell c_\ell^i = 1,$$

where $m_{t,i} \in \{0,1\}$ is a verifiability mask.

---

[2]MATH-VERIFY (Hugging Face), GitHub: huggingface/Math-Verify. We use it as a parsing/normalization front-end and then apply a numeric comparator.

**Reasoner local design.** Coefficients:

$$c_{\text{fmt}}^{\text{Reasoner}} = 0.20, \quad c_{\text{tool}}^{\text{Reasoner}} = 0.00, \quad c_{\text{step}}^{\text{Reasoner}} = 0.80.$$

Component scores (pure numerical check):

$$s_{\text{fmt},t}^{\text{Reasoner}} = \mathbf{1}\{\text{required output schema matched at turn } t\},$$

$$s_{\text{step},t}^{\text{Reasoner}} = \begin{cases} \text{NUMEQ}_\delta(\hat{y}, y^\star), & \text{if MATHVERIFIER extracts a numeric } \hat{y}, \\ 0, & \text{otherwise}, \end{cases}$$

and we do not use a tool-related step, so the corresponding score is implicitly $s_{\text{tool},t}^{\text{Reasoner}} \equiv 0$. The mask is

$$m_t^{\text{Reasoner}} = \mathbf{1}\{y^\star \text{ available (MATHVERIFIER) at turn } t\}.$$

For the Reasoner, $r_{t,\text{Reasoner}}^{\text{loc}}$ is obtained by plugging these coefficients, scores and the mask into the generic form above.

### B.2 CODE REWARD DESIGN

We consider code synthesis with unit tests. Let $\mathcal{S}$ be the active test suite and

$$p = \frac{1}{|\mathcal{S}|} \sum_{t \in \mathcal{S}} \mathbf{1}\{\text{RUN}(t, \text{code}) = \text{pass}\} \in [0, 1].$$

Team reward is dense; we again broadcast it over turns:

$$r_t^{\text{team}} = p, \qquad \forall t.$$

**Local rewards.** Each agent $i$ at turn $t$ uses a masked convex combination of verifiable sub-scores $s_{\ell,t}^i \in [0, 1]$:

$$r_{t,i}^{\text{loc}} = m_{t,i} \sum_\ell c_\ell^i s_{\ell,t}^i, \qquad \sum_\ell c_\ell^i = 1.$$

**Coder local reward.** We define the Coder's local reward as a weighted combination of basic sanity checks and the fraction of golden tests passed by the generated code. Coefficients:

$$c_{\text{build}}^{\text{Coder}} = 0.10, \quad c_{\text{run}}^{\text{Coder}} = 0.10, \quad c_{\text{nr}}^{\text{Coder}} = 0.80.$$

Let $\mathcal{T}^{\text{gold}}$ be the fixed set of golden unit tests for this problem. Component scores at turn $t$ are

$$s_{\text{build},t}^{\text{Coder}} = \mathbf{1}\{\text{the candidate code compiles/imports without syntax errors at } t\},$$

$$s_{\text{run},t}^{\text{Coder}} = \mathbf{1}\{\text{a smoke subset of } \mathcal{T}^{\text{gold}} \text{ runs without uncaught exceptions/timeout at } t\},$$

$$s_{\text{nr},t}^{\text{Coder}} = \frac{1}{|\mathcal{T}^{\text{gold}}|} \sum_{u \in \mathcal{T}^{\text{gold}}} \mathbf{1}\{\text{RUN}(u, \text{code}) = \text{pass}\},$$

i.e., the fraction of golden tests passed by the current code code. We apply an availability mask

$$m_t^{\text{Coder}} = \mathbf{1}\{\text{build/run logs and golden-test results are available at } t\},$$

and define the Coder's local reward as

$$r_{t,\text{Coder}}^{\text{loc}} = m_t^{\text{Coder}}\left(c_{\text{build}}^{\text{Coder}} s_{\text{build},t}^{\text{Coder}} + c_{\text{run}}^{\text{Coder}} s_{\text{run},t}^{\text{Coder}} + c_{\text{nr}}^{\text{Coder}} s_{\text{nr},t}^{\text{Coder}}\right).$$

**Tester local design.** Coefficients :

$$c_{\text{valid}}^{\text{Tester}} = 0.20, \quad c_{\text{nr}}^{\text{Tester}} = 0.80.$$

$$s_{\text{nr},t}^{\text{Tester}} = \frac{1}{|\mathcal{U}|} \sum_{u \in \mathcal{U}} \mathbf{1}\{\text{RUN}(u, \text{code}^\star) = \text{pass} > \tau_{mut}\},$$

where $\tau_{\text{mut}} = 0.60$.

Mask:

$$m_t^{\text{Tester}} = \mathbf{1}\{\text{test runner and mutation/coverage reports available at } t\}.$$

The Tester local reward $r_{t,\text{Tester}}^{\text{loc}}$ is obtained by combining $c_{\cdot}^{\text{Tester}}$, $s_{\cdot,t}^{\text{Tester}}$ and $m_t^{\text{Tester}}$ via the generic local-reward formula.

### B.3 SUDOKU REWARD DESIGN

We consider $N \times N$ Sudoku. Let $h_t$ be the answer action at turn $t$ and SOLVED$(\cdot)$ check row/column/subgrid validity. Team reward is a sparse success signal at termination, broadcast across turns:

$$r_t^{\text{team}} = \mathbf{1}\{\text{SOLVED}(h)=\text{true}\} \in \{0,1\}, \qquad \forall t.$$

**Local rewards.** Each agent $i$ at turn $t$ uses a masked convex combination of verifiable sub-scores $s_{\ell,t}^i \in [0,1]$:

$$r_{t,i}^{\text{loc}} = m_{t,i} \sum_\ell c_\ell^i s_{\ell,t}^i, \qquad \sum_\ell c_\ell^i = 1.$$

**Reasoner local design.** Coefficients :

$$c_{\text{fmt}}^{\text{Reasoner}} = 0.1, \quad c_{\text{legal}}^{\text{Reasoner}} = 0.1, \quad c_{\text{prog}}^{\text{Reasoner}} = 0.80.$$

Component scores (let $G_t$ be the current grid, $G_{t-1}$ the previous grid; 0 denotes empty):

$$s_{\text{fmt},t}^{\text{Reasoner}} = \mathbf{1}\{\text{action format is valid (full } N \times N \text{ grid or list of } [r,c,v])\},$$

$$s_{\text{legal},t}^{\text{Reasoner}} = \mathbf{1}\{\text{no row/column/subgrid duplicates in } G_t\},$$

$$s_{\text{prog},t}^{\text{Reasoner}} = \frac{1}{N^2} \sum_{r,c} \mathbf{1}\{G_{t-1}[r,c]=0,\ G_t[r,c]\neq 0\}.$$

Mask:

$$m_t^{\text{Reasoner}} = \mathbf{1}\{\text{we can parse the action and compute legality/progress at } t\}.$$

**Tool (executor) local design.** Coefficients (fixed):

$$c_{\text{fmt}}^{\text{Tool}} = 0.10, \quad c_{\text{exec}}^{\text{Tool}} = 0.10, \quad c_{\text{san}}^{\text{Tool}} = 0.80.$$

Component scores:

$$s_{\text{fmt},t}^{\text{Tool}} = \mathbf{1}\{\text{API/schema valid; values in } [1,N];\ \text{indices in bounds}\},$$

$$s_{\text{exec},t}^{\text{Tool}} = \mathbf{1}\{\text{no runtime error/timeout when applying edits}\},$$

$$s_{\text{san},t}^{\text{Tool}} = \begin{cases} 1, & \text{if all applied edits satisfy local Sudoku constraints,} \\ 0, & \text{otherwise.} \end{cases}$$

Mask:

$$m_t^{\text{Tool}} = \mathbf{1}\{\text{executor logs available and legality checks computed at } t\}.$$

### B.4 PLAN-PATH REWARD DESIGN

We consider 2D grid path planning on a $H \times W$ map with horizon $T$ and four-neighborhood moves. Let $d_t$ be the Manhattan distance from the current position to the goal at turn $t$ and $d_0 = \max(1, \text{initial distance})$ for normalization. Team reward is dense and distance-improving:

$$r_t^{\text{team}} = \begin{cases} 1, & \text{if at goal at } t, \\ \max(0,\ (d_{t-1} - d_t)/d_0), & \text{otherwise.} \end{cases}$$

Local rewards are masked convex combinations

$$r_{t,i}^{\text{loc}} = m_{t,i} \sum_\ell c_\ell^i s_{\ell,t}^i, \qquad \sum_\ell c_\ell^i = 1.$$

**Planner local design.** Coefficients (fixed):

$$c_{\text{fmt}}^{\text{Planner}} = 0.10, \quad c_{\text{leg}}^{\text{Planner}} = 0.10, \quad c_{\text{sp}}^{\text{Planner}} = 0.80.$$

Component scores at turn $t$ (action $a_t \in \{U, D, L, R\}$; $\mathcal{N}$ denotes passable neighbors; SPNEXT is 1 if $a_t$ lies on at least one shortest path from $s_{t-1}$ to goal, else 0):

$$s_{\text{fmt},t}^{\text{Planner}} = \mathbf{1}\{a_t \in \{U, D, L, R\}\},$$

$$s_{\text{leg},t}^{\text{Planner}} = \mathbf{1}\{\text{next cell in-bounds and not a wall}\},$$

$$s_{\text{sp},t}^{\text{Planner}} = \begin{cases} 1, & \text{if SPNEXT}(a_t){=}1, \\ 0, & \text{otherwise.} \end{cases}$$

Mask:

$$m_t^{\text{Planner}} = \mathbf{1}\{\text{map known and shortest-path oracle available at } t\}.$$

**Tool (executor/simulator) local design.** Coefficients:

$$c_{\text{fmt}}^{\text{Tool}} = 0.10, \quad c_{\text{exec}}^{\text{Tool}} = 0.10, \quad c_{\text{shape}}^{\text{Tool}} = 0.80.$$

Component scores (let $\phi_t = -d_t$ be the potential used in shaping):

$$s_{\text{fmt},t}^{\text{Tool}} = \mathbf{1}\{\text{action list parsable as } [\text{``U''}, \text{``D''}, \text{``L''}, \text{``R''}]\},$$

$$s_{\text{exec},t}^{\text{Tool}} = \mathbf{1}\{\text{no invalid move applied; simulation advances}\},$$

$$s_{\text{shape},t}^{\text{Tool}} = \mathbf{1}\{\phi_t \geq \phi_{t-1}\},$$

i.e., the potential does not decrease. Mask:

$$m_t^{\text{Tool}} = \mathbf{1}\{\text{execution logs and potentials } (\phi_{t-1}, \phi_t) \text{ available}\}.$$

### B.5 SOKOBAN REWARD DESIGN

We consider Sokoban with horizon $T$ on a fixed grid. Let $B$ be the number of boxes and $b_t$ the number of boxes on goal at turn $t$. Team reward is dense in box-on-goal ratio with terminal success at completion:

$$r_t^{\text{team}} = \begin{cases} 1, & \text{if all boxes on goals at } t, \\ b_t/B, & \text{otherwise.} \end{cases}$$

Local rewards are masked convex combinations

$$r_{t,i}^{\text{loc}} = m_{t,i} \sum_\ell c_\ell^i s_{\ell,t}^i, \qquad \sum_\ell c_\ell^i = 1.$$

**Planner local design.** Coefficients (fixed):

$$c_{\text{fmt}}^{\text{Planner}} = 0.10, \quad c_{\text{leg}}^{\text{Planner}} = 0.10, \quad c_{\text{dlk}}^{\text{Planner}} = 0.80.$$

Component scores at turn $t$ (action $a_t \in \{U, D, L, R\}$; PUSHOK $= 1$ if a planned push does not collide and stays in-bounds; DEADLOCKFREE $= 1$ if the move avoids standard static corner deadlocks for boxes not on goals):

$$s_{\text{fmt},t}^{\text{Planner}} = \mathbf{1}\{a_t \in \{U, D, L, R\}\},$$

$$s_{\text{leg},t}^{\text{Planner}} = \mathbf{1}\{\text{step is in-bounds and not into wall; if pushing, PUSHOK} = 1\},$$

$$s_{\text{dlk},t}^{\text{Planner}} = \begin{cases} 1, & \text{if DEADLOCKFREE} = 1, \\ 0, & \text{otherwise.} \end{cases}$$

Mask:

$$m_t^{\text{Planner}} = \mathbf{1}\{\text{grid known and deadlock heuristics evaluable at } t\}.$$

**Tool (executor/simulator) local design.** Coefficients (fixed):

$$c_{\text{fmt}}^{\text{Tool}} = 0.10, \quad c_{\text{exec}}^{\text{Tool}} = 0.10, \quad c_{\text{pot}}^{\text{Tool}} = 0.80.$$

Let $\psi_t = -\sum_{x \in \text{boxes}} \min_{g \in \text{goals}} \big(|x_r - g_r| + |x_c - g_c|\big)$ be the box-to-goal potential (larger is better). Component scores:

$$s_{\text{fmt},t}^{\text{Tool}} = \mathbf{1}\{\text{action list parsable; symbols match } \{\text{U}, \text{D}, \text{L}, \text{R}\}\},$$

$$s_{\text{exec},t}^{\text{Tool}} = \mathbf{1}\{\text{no illegal push; no wall/box collision}\},$$

$$s_{\text{pot},t}^{\text{Tool}} = \mathbf{1}\{\psi_t \geq \psi_{t-1}\}.$$

Mask:

$$m_t^{\text{Tool}} = \mathbf{1}\{\text{execution logs and potentials } (\psi_{t-1}, \psi_t) \text{ available}\}.$$

## B.6 OUTCOME-ONLY REWARD DESIGN

The shaped rewards in Sections B 1–5 provide rich, task-specific feedback (e.g., shortest-path signals in Plan-Path and deadlock heuristics in Sokoban). To isolate the contribution of such complex shaping from that of the AT-GRPO algorithm itself, we additionally consider a simplified **outcome-only** reward design used in our ablation studies.

The **team reward** is strictly binary and episodic. Let $\mathbb{I}(\text{Success})$ denote the environment success indicator. The team reward is defined and broadcast over turns as

$$r_t^{\text{team}} = \mathbb{I}(\text{Success}), \qquad \forall t.$$

The **per-agent local reward** in this setting is an auxiliary binary signal that only checks whether agent $i$ produced a validly formatted action (e.g., correct API call or JSON structure). Let $\mathbb{I}(\text{FmtValid}_t^i)$ denote the indicator that the output of agent $i$ at turn $t$ satisfies all formatting constraints. We define

$$r_{t,i}^{\text{loc}} = r_t^{i,\text{out}} = \mathbb{I}\big(\text{FmtValid}_t^i\big).$$

The final per-agent reward $r_{t,i}$ is then combined according to Eq. 3,

$$r_{t,i} = \alpha \, r_t^{\text{team}} + r_{t,i}^{\text{loc}},$$

where we use a fixed, task-independent $\alpha$ shared with the shaped-reward configurations. By utilizing sparse episodic rewards and simple formatting checks rather than dense shaping signals, this outcome-only configuration is significantly more general and provides a baseline for our algorithm ablation.

## B.7 THEORETICAL JUSTIFICATION FOR GREEDY TURN-LEVEL TRANSITIONS

In this section, we formally justify the optimality of greedy selection based on environment-verified rewards. Consider the underlying MDP with optimal action-value function $Q^*(s, a)$. The Bellman optimality principle implies that any policy $\pi^*$ satisfying $\pi^*(s) \in \arg\max_a Q^*(s, a)$ is optimal (Sutton and Barto, 2018).

We operate in a setting where the environment returns an *outcome-based verifiable reward*, $r_{\text{ver}}(s, a)$, for each action. We posit that this reward acts as a monotonic proxy for the true value function $Q^*(s, a)$: a higher verification score directly corresponds to a higher probability of final success. Consequently, maximizing the immediate verifiable reward is structurally equivalent to maximizing the long-term optimal value. We formalize this alignment as follows:

**Assumption 1** (Monotonicity of Verification Feedback). *For any state $s$ and actions $a_1, a_2$, the verifiable reward preserves the ordering of the optimal action-value function:*

$$r_{\text{ver}}(s, a_1) > r_{\text{ver}}(s, a_2) \implies Q^*(s, a_1) \geq Q^*(s, a_2). \tag{4}$$

*This implies that $r_{\text{ver}}(s, \cdot)$ and $Q^*(s, \cdot)$ induce consistent rankings over the action space at any state $s$.*

**Lemma 1** (Equivalence of Maximizers). *Under Assumption 1, the set of actions maximizing the verifiable reward is a subset of the actions maximizing the optimal Q-function:*

$$\arg\max_a r_{\text{ver}}(s, a) \subseteq \arg\max_a Q^*(s, a). \tag{5}$$

*Proof.* Let $a^*_{\text{ver}} \in \arg\max_a r_{\text{ver}}(s, a)$. Suppose, for the sake of contradiction, that there exists an action $a'$ such that $Q^*(s, a') > Q^*(s, a^*_{\text{ver}})$. By the contrapositive of Assumption 1, strict inequality in $Q^*$ implies strict inequality in $r_{\text{ver}}$ (given consistent rankings), which would imply $r_{\text{ver}}(s, a') \geq r_{\text{ver}}(s, a^*_{\text{ver}})$. Since $a^*_{\text{ver}}$ is a maximizer, strict inequality is impossible. If equality holds, $a'$ is also a maximizer of $r_{\text{ver}}$, and by the consistency assumption, it must share the same optimal $Q$-value. Thus, any action maximizing $r_{\text{ver}}$ necessarily maximizes $Q^*$. □

**Proposition 1** (Optimality of Verifier-Greedy Policy). *Let $\pi_{\text{ver}}$ be a deterministic policy such that $\pi_{\text{ver}}(s) \in \arg\max_a r_{\text{ver}}(s, a)$ for all states s. Under Assumption 1, $\pi_{\text{ver}}$ is an optimal policy.*

*Proof.* By Lemma 1, selecting an action that maximizes the immediate verification score ensures that $\pi_{\text{ver}}(s) \in \arg\max_a Q^*(s, a)$. Consequently, $\pi_{\text{ver}}$ satisfies the Bellman optimality equation at every state. □

In our implementation, we approximate this policy by sampling candidate actions and greedily selecting the one with the highest $r_{\text{ver}}$. Proposition 1 guarantees that this strategy effectively performs a greedy search over the support of sampled actions with respect to the true optimal value function $Q^*$, avoiding the myopic bias typically associated with greedy transitions.

## C  EXPERIMENT DETAILS

### C.1  TRAINING DETAILS

All methods share the same hyperparameters unless noted. The maximum response length is **4096** tokens, and the (task-specific) maximum prompt length is set to accommodate turn-by-turn dialogue history: **8192** tokens for *mathematics* and *code* tasks, and **16384** tokens for all other symbolic tasks. Training uses a global batch size of **128**, with **PPO mini-batch size 64** and gradient clipping at **1.0**. The actor is optimized with Adam at a learning rate of **1e-6** and weight decay **0.01**. We adopt **GRPO** for advantage estimation with $\gamma$=1.0 and $\lambda$=1.0. Entropy regularization is off ($\text{entropy\_coeff}$=0). The sample temperature $T_{sample} = 1.0$, top-$p$=1.0, top-$k$=$-1$, and 4 sample per prompt; validation is deterministic (temperature **0**, $\text{do\_sample}$=**False**). rewards are computed by a rule-based function ($\text{compute\_score}$) when provided. Both models are trained for 150 steps.

### C.2  PROMPT DESIGN

**Code MAS Workflow**

PHASE 1: GENERATION

In the initial phase, both agents are given a problem description. The Coder is prompted to generate a solution, while the Tester is prompted to generate a corresponding test case.

---

**Code Agent (Coder): Turn 0**

**Input:**

- **Problem:** A natural language description of a programming task.

**Prompt:**
You are a helpful assistant that writes Python to solve the problem. Think step by step, then output code. Important: - Read all inputs via input(). - Print all results with print(). - Do not hardcode or fabricate inputs. Now solve: Problem: "'problem description'" First, decide on the number and types of inputs required (e.g., x = int(input()), b = int(input())), then

---

implement the solution and print the result. Please answer in the following format: Code:
` ` `python (your code here)` ` `
**Output:** Code

---

**Unit Tester Agent (Test-Case Author): Turn 0**

**Input:**

- **Problem:** A natural language description of a programming task, e.g., {problem}.

**Prompt:**
You are a helpful assistant that creates unit test cases (input + expected output) for a coding task.
Problem: "' problem discrption"'
Provide one new high-quality test case. Before giving the test case, reason carefully to ensure the output is correct, then derive the output for your chosen input. Respond in the format:
**Test Input:**` ` `input here` ` ` **Test Output:**` ` `output here` ` `
**Output:** Test input, Test Output.

---

PHASE 2: REFINEMENT

In subsequent turns, the agents receive feedback based on mismatches between the generated code and test cases. They are prompted to refine their previous outputs.

---

**Code Agent (Coder): Turn > 0**

**Input:**

- **Problem:** The original problem description, {problem}.
- **Mismatch History:** A record of previous code, test inputs, expected outputs, and actual execution outputs, highlighting any differences, {mismatch_history}.

**Prompt:**
You are a helpful assistant that corrects and refines code.
Important: - Read inputs via input(); output with print(). - Do not hardcode inputs.
Problem: {problem}
Use the history below to guide your fixes:
{mismatch_history}
If your previous code crashed, first fix the bug.
If execution succeeded but outputs mismatched the expected output, decide if the test case is correct. - If the test is correct, refine your code to pass it. - If the test is wrong, verify your program's logic and keep it.
Provide the final, corrected code. Respond in the format:
Code: ` ` `python # your code here` ` `
**Output:** Code

---

**Unit Tester Agent (Test-Case Author): Turn > 0**

**Input:**

- **Problem:** The original problem description, {problem}.
- **Mismatch History:** A record showing the test case and the differing execution output from the Coder's program, {mismatch_history}.

**Prompt:** You are an assistant that checks and refines unit tests for a coding task.
Problem: problem
Analyze the history below:

> {mismatch_history}
> First, decide whether your previous test case was correct (watch for misunderstandings of the task). If it was wrong or unclear, provide a corrected test case. Respond in the format:
> **Test Input:**, **Test Output:**
> **Output:** Test input, test output.

**Math MAS Workflow**

PHASE 1: GENERATION

In the initial phase, two complementary agents are given the same math problem. The *Reasoning Agent* produces a step-by-step mathematical solution and a boxed final answer. The *Python Tool Agent* writes executable Python that computes (and prints) the final answer.

---

**Reasoning Agent: Turn 0**

**Input:**

- **Problem:** A mathematical problem in natural language.

**Prompt:**
You are a helpful assistant that solves math problems via careful reasoning.
Problem: problem
First, outline the key reasoning steps. Then carry out the full solution. After solving, present the final answer in a LaTeX box.
Before giving the full reasoning, summarize the steps clearly in: **Reasoning Steps:** 'reasoning steps here'
Then provide your complete solution concisely. Put your final answer in: ###
Rules:
* The boxed value must be a single number or expression (simplified if possible). * Do not add words after the box; only the final value goes after ####.
Output format:
1. Your reasoning (short and clear). 2. Final line must contain only the boxed answer, e.g., #### 123.
**Output:** Reasoning solution and a final answer after ####.

---

**Python Tool Agent (Coder for Math): Turn 0**

**Input:**

- **Problem:** The same mathematical problem, {problem}.

**Prompt:**
You are a helpful programming assistant that writes Python to solve the math problem.

**Problem**  {problem}

**Requirements**

- Write correct, readable Python that computes the final answer.
- Think step by step in comments if helpful.
- Use only the standard library and deterministic math (no internet, no randomness).
- At the end, PRINT ONLY the final numeric or symbolic answer (nothing else).

**Output:** Code (the program prints the final answer).

---

PHASE 2: REFINEMENT

From the second turn onward, agents receive feedback derived from mismatches between the Reasoning Agent's boxed answer and the Python Tool Agent's printed output. Each agent uses the history to refine its output.

---

**Reasoning Agent (Math Solver): Turn $> 0$**

**Input:**

- **Problem:** The original problem, {problem}.
- **Mismatch History:** Prior reasoning ({reasoning_solution}), its extracted answer ({reasoning_extracted_answer}), the Python code ({code_solution}), and the code's printed output ({code_extracted_answer}), summarized as {mismatch_history}.

**Prompt:**
You are a helpful assistant that refines mathematical solutions through reasoning.
Problem: problem
History (previous attempts and outputs): mismatch history
First, compare your previous boxed answer with the Python Tool Agent's printed output.
* If the code output corrects a computational slip in your reasoning, adopt the corrected value. * If the code likely has a bug (e.g., mishandled edge cases, precision, domains), keep the mathematically correct answer and explain briefly.
Then solve the problem again, more robustly.
Before giving the full reasoning, summarize the key steps clearly: **Reasoning Steps:** 'reasoning steps here'
Finish with the final answer after: ####
Final line must contain only the boxed value (no extra text).
**Output:** Updated reasoning and a final answer after ####.

---

**Python Tool Agent (Coder for Math): Turn $> 0$**

**Input:**

- **Problem:** The original problem, {problem}.
- **Mismatch History:** Prior code and printed output, and the Reasoning Agent's solution and boxed answer, summarized as {mismatch_history}.

**Prompt:**
You are a helpful programming assistant that refines Python solutions for math problems.
Problem: problem
History (reasoning vs. execution mismatches): mismatch history
Tasks:
1. Judge whether the Reasoning Agent's boxed answer or your previous printed result is more likely correct (consider numerical stability, edge cases, exact vs. float). 2. Fix or rewrite the code so it reliably computes the correct final answer.
* Prefer exact arithmetic (fractions, integers, rational simplification) when possible. * Add minimal checks for domain/edge cases. * Keep outputs deterministic.
Respond in the format:
**Code:**
```python # corrected code here # print ONLY the final answer on the last line ```
**Output:** Refined code (the program prints the final answer).

---

**Sudoku MAS Workflow**

In the initial phase, two complementary agents are given the same Sudoku-solving task on an $n \times n$ grid. The *Tool Agent* writes executable Python that outputs either a completed grid or a list of fill steps. The *Plan Agent* inspects the task, the tool code, and its execution output, then decides the final solution.

**Tool Agent (Sudoku Coder)**

**Input:**

- **Task Description:** {task}, including grid size, rules (rows/columns/sub-grids contain unique digits), and any constraints.
- **Env Context:** {env_context} (e.g., {size}, {subgrid_size}, {puzzle}, {observation}).

**Prompt:**
You are an AI assistant designed to be helpful. Utilize your programming expertise to address the task. Propose Python code (within a single python code block) for the user to run. Ensure each response contains only ONE code block. Use the 'print' function to output EITHER: (A) the completed grid as a JSON array of arrays, OR (B) a JSON list of fill steps (r,c,v) using 1-based indices.
Formatting requirements:
* The program's output is the Sudoku solution: eg: [[5,3,4,6,7,8,9,1,2], ..., [3,4,5,2,8,6,1,7,9]]
* Print ONLY the JSON (no extra text, no comments).
Task: Solve the sizexsize Sudoku. Fill digits 1..size ; rows, columns, and sub-grids must have unique digits.
Current puzzle (dots denote blanks): observation
Environment: - size - subgrid_size: subgrid_size - notes/constraints: constraints
**Output:** Code (program prints either the completed grid JSON or a JSON list of fill steps).

**Plan Agent (Planner & Verifier)**

**Input:**

- **Task Description:** {task}.
- **Tool Code:** {tool_code}.
- **Tool Execution Output:** {tool_execution_output}.
- **Tool Proposed Solution:** {tool_solution} (JSON grid or JSON steps).
- **Observation (for reference):** {observation}.

**Prompt:** You are a planning and reasoning agent. You will receive:
* The original task description * The Tool Agent's code * The code execution output (a JSON grid or JSON steps)
Your job is to reason carefully, decide the final Sudoku solution, and format your response EXACTLY as specified.
Instructions:
* Read the task, inspect the code, and verify the execution output against the Sudoku rules: rows, columns, and sub-grids must contain unique digits in 1..n. * If the tool's output is a complete, valid solution, adopt it. * If it is incomplete or violates constraints, correct it or provide your own. * Keep reasoning concise but explicit: explain why the final result is valid.
FORMATTING IS MANDATORY. Give the final answer AFTER the line that begins with ####. You may return EITHER: - a completed grid as JSON, OR - a JSON list of fill steps (r,c,v), 1-based indices.
Examples:
#### [[5,3,4,6,7,8,9,1,2], ..., [3,4,5,2,8,6,1,7,9]]
#### [[1,3,4],[2,1,6],[9,9,1]]
**Output:** Final Sudoku answer (completed grid JSON or JSON steps).

## D PLAN-PATH MAS WORKFLOW

### PHASE 1: GENERATION

In the initial phase, two complementary agents are given the same path-planning task on a grid/world. The *Tool Agent* writes executable Python that outputs an action list (e.g., $[U, R, D, L]$). The *Plan Agent* inspects the task, the tool code, and its execution output, then decides the final action list.

---

**Tool Agent (Path Coder): Turn 0**

**Input:**

- **Task Description:** {task}, including grid/map, start, goal, obstacles, and constraints.
- **Env Context:** {env_context} (e.g., {grid}, {start}, {goal}, {obstacles}, {constraints}).

**Prompt:**
You are an AI assistant designed to be helpful. Utilize your programming expertise to address the task. Propose Python code (within a single python code block) for the user to run. Ensure each response contains only ONE code block. Use the 'print' function to output the action list that moves from the start to the goal. You may output the full action list if you can reach the target, or a partial list if uncertain.
Formatting requirements:
* Begin the Python block with `python and end with `. * The program's output IS the action list (e.g., [U,R,D,L]). * Print ONLY the action list (no extra text).
Task: task
Environment: env_context
**Output:** Code (program prints an action list, e.g., [U,R,D,L]).

---

**Plan Agent (Planner & Verifier): Turn 0**

**Input:**

- **Task Description:** {task}.
- **Tool Code:** {tool_code}.
- **Tool Execution Output:** {tool_execution_output}.
- **Tool Proposed Action:** {tool_action}.

**Prompt:**
You are a planning and reasoning agent. You will receive:
* The original task description * The Code Agent's (Tool Agent's) code * The code execution output
Your job is to reason carefully, decide the final action list, and format your response EXACTLY as specified.
Instructions:
* Read the task, inspect the code, and verify the execution output against the task requirements and environment constraints (bounds, obstacles, goal). * If the code/output is correct and sufficient, adopt it. * Otherwise, improve or override it with your own reasoning. * Keep reasoning concise but explicit: justify why the final action is correct.
FORMATTING IS MANDATORY. Give the final action list AFTER the line that begins with ####. Example:
#### [U,R,D,L]
**Output:** Final action list.

---

PHASE 2: REFINEMENT

From the second turn onward, agents receive feedback based on mismatches between the Tool Agent's printed action list and feasibility checks from the environment or the Plan Agent's assessment. Each agent uses the history to refine its output.

---

**Tool Agent (Path Coder): Turn $> 0$**

**Input:**

- **Task Description:** {task}.
- **Mismatch/Trajectory History:** Prior code and printed actions, planner feedback, and (action, state) pairs, summarized as {action_state_history}.

**Prompt:**
Refine your Python solution to produce a correct, executable action list.
Task: task
History (previous attempts, planner feedback, and trajectory): action_state_history
Requirements:
* Output must be an action list that reaches the goal without violating constraints (stay in-bounds, avoid obstacles). * If certain, print the full list; if uncertain, print a safe partial prefix. * Single python code block only; program output IS the action list. * Begin with 'python and end with '; print ONLY the action list (e.g., [U,R,D,L]).
Respond in the format:
**Code:**
``` python
# corrected code here # last line prints ONLY the action list
```

**Output:** Refined code (program prints an action list).

---

**Plan Agent (Planner & Verifier): Turn $> 0$**

**Input:**

- **Task Description:** {task}.
- **Tool Code:** {tool_code}.
- **Tool Execution Output:** {tool_execution_output}.
- **Tool Proposed Action:** {tool_action}.
- **Action State History (if any):** For each step $i$, The $i$-th action is $a_i$. The $i$-th state is $s_i$. Summarized as action state history.

**Prompt:**
You are a planning and reasoning agent.
Task: task
Tool Agent's latest code and output:
* Code: tool_code * Execution output: tool_execution_output * Proposed action: tool_action
Trajectory/history: action_state_history
Instructions:
* Verify feasibility of the proposed action sequence step by step. * If it collides, goes out of bounds, loops, or fails to reach the goal, correct it (you may shorten, extend, or replace the sequence). * Prefer the simplest valid plan; if uncertain, provide the best safe prefix and explain briefly. * Keep reasoning concise but explicit.
FORMATTING IS MANDATORY. Give the FINAL action list AFTER the line that begins with ####. Example:
#### [U,R,D,L]
**Output:** Final action list.

## E    ABLATION STUDY OF OUTCOME REWARD

Table 6: **Performance Comparison with Sparse Outcome-Only Rewards.** To address concerns regarding reward engineering, we evaluate AT-GRPO using only sparse outcome signals (Outcome-only), removing all intermediate heuristics. Even without dense guidance, our method maintains high performance and significantly outperforms the baselines.

| Task | Baselines | | AT-GRPO (Ours) | | Robustness |
|------|-----------|------|--------------|-----------------|-------------|
| | SA | MAS | **Outcome-only** | Dense (Original) | (Drop $\Delta$) |
| Sokoban | 48.0% | 72.0% | **93.0%** | 96.0% | -3.0% |
| Sudoku | 9.0% | 16.0% | **99.5%** | 99.5% | **0.0%** |
| Plan-Path | 12.0% | 71.0% | **89.0%** | 93.0% | -4.0% |

A potential concern with the dense task-specific rewards (detailed in Appendix B) is that they might provide "oracle" guidance (e.g., distance-to-goal heuristics in Plan-Path), thereby simplifying the reasoning challenge. To disentangle the contribution of the AT-GRPO algorithm from the reward design, we evaluate our method using the **Outcome-only** reward formulation defined in Appendix B.6. In this setting, all intermediate heuristic signals are removed, and the agents receive positive feedback only upon successfully solving the final task, exactly matching the sparse signal availability of the baselines.

Table 6 compares the performance of AT-GRPO under dense versus sparse outcome-only rewards against the SA and MAS baselines. We observe two key findings:

1. **Independence from Dense Heuristics:** The removal of dense rewards results in only marginal performance degradation. For instance, on the *Plan-Path* task—where the dense reward provided shortest-path information—the accuracy drops by only 4.0% (from 93.0% to 89.0%). On *Sudoku*, the performance remains identical at 99.5%. This indicates that while dense rewards accelerate learning, they are not a prerequisite for the model's success.

2. **Superiority over Baselines:** Even in the sparse outcome-only setting, AT-GRPO maintains a decisive advantage over the baselines. On *Plan-Path*, our sparse-reward performance (89.0%) vastly outperforms the SA baseline (12.0%) and the MAS baseline (71.0%). This dramatic gap (+77% vs. SA) under identical reward conditions strongly refutes the hypothesis that our results are confounded by reward engineering. Instead, it demonstrates that the cooperative group optimization mechanism is intrinsically capable of solving complex planning tasks without oracle guidance.

## F    MULTI TURN SINGLE AGENT

Table 7: Single-agent ablations on **Code** and **Math** (Qwen3 1.7B).

| Setting | Code | | | Math | | |
|---------|------|------|-------------|------|--------|----------|
| | LiveCodeBench | APPS | CodeContests | AIME24 | AIME25 | Olympiad |
| SA, single turn | 11.6 | 16.2 | 3.6 | 13.4 | 9.8 | 22.2 |
| SA + multi-turn | 10.4 | 10.4 | 0.0 | 3.3 | 6.7 | 15.8 |
| SA, single turn + RL | 18.8 | 17.0 | 3.0 | 10.0 | 6.7 | 23.8 |
| SA, multi-turn +RL | 17.7 | 13.3 | 1.2 | 6.67 | 3.3 | 16.9 |

**On the effectiveness of multi-turn single-agent variants.**    Tab. 7 and Tab. 8 report single-agent ablations on Code and Math. For both 1.7B and 8B models, introducing a multi-turn SA variant (i.e., letting one agent repeatedly revise its own answer) brings no consistent benefit over the standard single-turn SA baseline and often degrades performance, which is align with the obeservation in Chen et al. (2025b) . For example, at the SFT stage on Qwen3-1.7B, LiveCodeBench drops from 11.6 to 10.4 and AIME24 from 13.4 to 3.3 when switching from single-turn SA to multi-turn SA, with similar trends on AIME25 and Olympiad. After RL, the single-turn SA policy still outperforms

Table 8: Single-agent ablations on **Code** and **Math** (Qwen3 8B).

| Setting | Code | | | Math | | |
|---|---|---|---|---|---|---|
| | LiveCodeBench | APPS | CodeContests | AIME24 | AIME25 | Olympiad |
| SA, single turn | 22.8 | 30.2 | 15.75 | 18.3 | 20.0 | 55.0 |
| SA + multi-turn | 7.8 | 20.3 | 5.12 | 16.7 | 16.7 | 53.4 |
| SA, single turn + RL | 25.7 | 37.0 | 12.12 | 18.3 | 26.67 | 54.8 |
| SA, multi-turn + RL | 16.8 | 35.4 | 11.1 | 16.7 | 23.3 | 51.2 |

its multi-turn counterpart across most Code and Math benchmarks for both model scales. These results support our claim in the main text: in the absence of additional environmental signals or feedback from complementary roles, multi-turn SA interaction is a contrived use of extra turns that departs from the QA-style pretraining regime and fails to translate into improved task performance, in contrast to our multi-agent workflows where multi-turn interaction with structured cross-agent feedback yields clear gains.

## G  SYSTEM COMPLEXITY OF AGENT- AND TURN-WISE GROUPING

Our on-policy RL framework operates by alternating between two distinct phases: **inference** (rollout generation) and **training** (loss computation and parameter updates). In this section, we analyze the computational and memory complexity of AT-GRPO (Alg. 1) and discuss how it scales with the number of agents and the turn horizon under different MAS interaction patterns and system constraints.

**Notation.** Let $N$ be the number of agents, $T$ the turn horizon, $E$ the number of parallel environment instances, and $K$ the sampling factor (number of candidate actions per agent–turn in tree sampling). Let $L$ denote the average number of generated tokens per action.

### G.1  INFERENCE TIME COMPLEXITY

#### G.1.1  SYSTEM DESIGN WITH ASYNCHRONOUS VLLM GENERATION

Our implementation uses a vLLM-style *asynchronous* engine with continuous batching for both rollouts and evaluation: each agent–turn query $(e, i, t)$ is submitted as an independent request, and the engine maintains a token-level scheduler that dynamically adds new sequences and removes finished ones. Compared to a naive synchronous design that forms a fixed batch of agent responses and waits for the longest sequence to finish, this asynchronous scheme largely eliminates long-tail stragglers, keeps the GPUs close to saturation, and naturally interleaves agent–turns from parallel and sequential MAS workflows into efficient pipeline.

#### G.1.2  INFERENCE-TIME COMPLEXITY

During inference ($K{=}1$), we analyze the per-episode wall-clock latency. The complexity depends on the execution schedule—Sequential or Parallel—determined by the interaction logic. Crucially, the baseline Single-Agent (SA) complexity also varies by task: for **Code** and **Math**, the SA baseline is typically single-turn ($T{=}1$), whereas for **Plan** and **Game**, the SA baseline involves multi-turn interactions ($T{>}1$). We denote the baseline latency as $\text{Time}_{\text{infer}}^{\text{SA}}$.

**Sequential MAS.** In this setting (e.g., **Plan**, **Game**), agents act serially within each turn to condition on updated history. While the SA baseline requires $T$ sequential steps, the Sequential MAS requires $N$ serial agent moves for each of the $T$ turns, resulting in a critical path of $N \cdot T$. Comparing this to the multi-turn SA baseline:

$$\frac{\text{Time}_{\text{infer}}^{\text{Seq}}}{\text{Time}_{\text{infer}}^{\text{SA}}} \;\leq\; \frac{N \cdot T}{T} \;=\; N.$$

Thus, the latency overhead scales linearly with the number of agents $N$.

**Parallel MAS.** In this setting (e.g., **Code**, **Math**), we employ multi-round debate where all $N$ agents act in parallel in each round. By leveraging continuous batching, the $N$ concurrent queries are processed together on the inference engine. However, the parallelism is not unbounded: for a fixed model and hardware budget, there exists a maximum number of concurrent sequences the engine can hold in memory.

Let $B_{\max}$ denote the maximum number of concurrent sequences that can be served by the cluster (determined by GPU memory, model size, and the target context length). With $E$ parallel environments and $K$ candidates per agent (e.g., $K$ GRPO samples), the number of sequences per MAS step is $E \cdot N \cdot K$. To keep all agents truly parallel, we must satisfy

$$E \cdot N \cdot K \;\leq\; B_{\max}.$$

Equivalently, the maximum parallelizable agent count is

$$N_{\max} \;=\; \left\lfloor \frac{B_{\max}}{E \cdot K} \right\rfloor.$$

When $N \leq N_{\max}$, the $N$ agents at each turn can be fully batched, and the latency scales primarily with the debate depth $T$:

$$\frac{\text{Time}_{\text{infer}}^{\text{Para}}}{\text{Time}_{\text{infer}}^{\text{SA}}} \;\lesssim\; T.$$

When $N > N_{\max}$, the engine automatically schedules the $N$ agents in $\lceil N/N_{\max} \rceil$ waves, and the latency bound becomes

$$\frac{\text{Time}_{\text{infer}}^{\text{Para}}}{\text{Time}_{\text{infer}}^{\text{SA}}} \;\lesssim\; T \cdot \lceil N/N_{\max} \rceil,$$

which smoothly reduces to the single-wave case when $N \leq N_{\max}$.

## G.2 TRAINING-TIME COMPLEXITY

The computational bottleneck during training lies in the forward and backward passes for the collected candidate actions. For $E$ environments, $N$ agents, and $T$ turns, with $K$ samples each, the total rollout sample size is $|\mathcal{D}|_{\text{MAS}} = E \cdot N \cdot T \cdot K$.

The proposed agent- and turn-wise grouping introduces only **a lightweight hashing overhead** of $O(|\mathcal{D}|_{\text{MAS}})$, which is negligible compared to the token-level model execution $O(|\mathcal{D}|_{\text{MAS}} \cdot L \cdot C_{\text{model}})$. Therefore, the complexity relationship between our multi-agent approach and the standard single-agent GRPO ($|\mathcal{D}|_{\text{SA}} = E \cdot T \cdot K$) is defined by the ratio of their rollout sample sizes:

$$\frac{\text{Time}_{\text{train}}^{\text{MAS}}}{\text{Time}_{\text{train}}^{\text{SA}}} \;\leq\; \frac{|\mathcal{D}|_{\text{MAS}}}{|\mathcal{D}|_{\text{SA}}} \;=\; NT.$$

This demonstrates that our method introduces no extra asymptotic complexity beyond a linear scaling with the number of agents $N$.

## G.3 EMPIRICAL LATENCY STUDY

We conducted latency profiling on a cluster of four H100 GPUs with an effective decoding batch size of $32 \times 8$. For the **Code** task, one on-policy iteration for the single-agent baseline ($N{=}1, T{=}1$) requires approximately 4 minutes for rollout (inference) and 1 minute for AT-GRPO training; thus, inference dominates roughly 80% of the total wall-clock time. Scaling to the MAS setting ($N{=}2$, multi-turn) approximately results in 8 minutes for rollout and 2 minutes for training.

In the **Game** domain, while training costs remain comparable to the Code task, we observe an inversion in inference latency. The single-agent baseline averages 2.8 minutes per rollout, whereas the MAS inference time drops to 1.5 minutes. This reduction is attributable to the superior performance of MAS: the group efficiently completes tasks in fewer turns (triggering early termination), whereas the single-agent policy frequently struggles and exhausts the maximum turn horizon.

# H   CASE STUDIES OF MAS WORKFLOWS

This appendix presents two concrete multi-agent case studies, one in a box-pushing grid game and one in code generation with unit tests. For each domain, we include the original prompts and agent-facing messages, and we distinguish erroneous behaviors from successful ones using × and ✓, respectively.

## H.1   MAS FOR GAME

**Task.**   Task: Planner proposes the next action sequence; Executor calls environment tools (simulator, legality checker, shortest-path/BFS helper) to apply actions and return effects/observations (updated grid, agent/box poses, success/failure flags). Episode ends when the goal is met (all boxes on targets) or the turn budget is reached.

**Before RL (×).**   Before RL: The Plan Agent gets a valid path for the box from Tool agent but completely misses the point. It tries to follow the box's path itself, runs straight into a wall, and fails instantly. It doesn't understand its job is to push the box, not be the box.

**After on-policy RL in MAS (✓).**   After on-policy RL in MAS: RL teaches the agent the difference. It learns that rewards come from moving the box along the designated path. This insight forces it to discover the correct low-level strategy: first, navigate behind the box, then execute the push.

---

**The Puzzle (Game Grid)**

**Input (environment prompt):**

```
###########
# . . . . . . G . #
# .### . . .# . #
# . # . . . . # . #
# . # . . . . # . #
# . # . B . . . # . #
#A . . . . . . # . #
###########
```

# wall, . free, A agent, B box, G goal

Start: A(8,1), B(6,6), G(1,8) — 1-indexed (row,col).
**Output:** Initial grid state for the MAS.

---

**Tool Agent (Box Path via BFS)**

**Input:**
- **Task Description:** Given the grid and object positions, compute an optimal path for the *box* from its start to the goal.

**Prompt (Tool Agent behavior):**
A tool agent perceives obstacles and calculates an optimal path for the **box** using BFS:
**Output (Tool Agent response):**

Box path: [R, R, U, U, U, U, R]
(Validated: free behind-space each step)

---

**Plan Agent × — Failed Interpretation**

**Input:**
- **Task Description:** Use the Tool Agent's box path to control the robot.

---

- **Tool Output:** [R, R, U, U, U, U, R].

**Prompt (narrative as shown to the agent):**
The planning agent directly applies the box path to itself, failing to understand that pushing a box is different from moving into a free space.
**Response (Plan Agent thought and action list):**

**Thought:**
*"Great, I'll execute these moves."*

**Planned robot actions:**
[R, R, U, U, U, U, R]

**Outcome:** Blocked by walls; never gets behind the box to push it. ⇒ **Failure** ×.

---

### Plan Agent ✓ — Correct Interpretation

**Input:**

- **Task Description:** Move the box from its start to the goal using the Tool Agent's plan.
- **Tool Output:** [R, R, U, U, U, U, R].

**Prompt (narrative as shown to the agent):**
The agent correctly translates the high-level plan (move the box) into a sequence of agent-specific actions (get behind the box, then push).
**Response (Plan Agent thought process):**

**Thought Process:**

1. Translate each box push to a required pre-push robot cell.

   - Push Right → Robot must be Left of box.
   - Push Up → Robot must be Below box.
   - (And so on for L and D...)

2. For each step in the box plan, route the agent to the corresponding pre-push cell (avoiding walls), then move one final tile in the push direction.

**Output:** A valid sequence of robot moves that first routes behind the box and then pushes it along the planned path, achieving the goal ✓.

---

## H.2 MAS FOR CODE

**Code Generation Case Study.** Task: Coder writes a solution; Unit-Tester writes tests. Terminate=all tests pass. Otherwise: each agent revises its own previous output using the environment feedback/results (Coder fixes code; Unit-Tester fixes unit test), then re-run.

**Before RL** (×). Before RL: The Plan Agent gets a valid path for the box from Tool agent but completely misses the point. It tries to follow the box's path itself, runs straight into a wall, and fails instantly. It doesn't understand its job is to push the box, not be the box.

**After on-policy RL in MAS** (✓). After on-policy RL in MAS: RL teaches the agent the difference. It learns that rewards come from moving the box along the designated path. This insight forces it to discover the correct low-level strategy: first, navigate behind the box, then execute the push.

---

### The Prompt (Unit-Test Generation)

**Input (environment prompt):**

A request is sent to a unit test generator to calculate the "prettiness" sum of an array a of size n using the formula:

$$\sum_{i=1}^{n} \sum_{j=1}^{n} \gcd(a_i, a_j) \cdot \gcd(i, j).$$

The generator is instructed to produce the test input and output following a specific format:

**Test Input:** input here
**Test Output:** output here

**Output:** Problem specification and required I/O format for the Unit-Tester.

---

### Unit-Tester × — Failure 1: Missing Output

**Prompt (failure description):**
The generator only provided the test input, failing to include the required output.
**Response (incorrect test case):**

**Test Input:**
n = 5
a = [3, 6, 2, 1, 4]

**Test Output:** (is missing)

**Outcome:** The required **Test Output:** field is missing entirely, so the test case is unusable ×.

---

### Unit-Tester × — Failure 2: Wrong Format

**Prompt (failure description):**
The output doesn't adhere to the specified **Test...** bolded format.
**Response (incorrect test case):**

test˙input:
n = 5
a = [3, 6, 2, 1, 4]

test˙output:
77

**Outcome:** Although both input and output are present, the format violates the required bolded **Test Input:** and **Test Output:** structure, so the checker rejects it ×.

---

### Unit-Tester ✓ — Success (MAS + AT-GRPO)

**Prompt (success description):**
Correct Case (MAS + AT-GRPO)
The output correctly follows the format, provides both input and output, and ensures all variables are consistent.
**Response (correct test case):**

**Test Input:**
n = 5
a = [3, 6, 2, 1, 4]

**Test Output:**
77

**Outcome:** The test fully respects the prescribed format and contains consistent input and output fields, enabling reliable automatic checking ✓.

