# OpenReview forum: "Stronger-MAS: Multi-Agent Reinforcement Learning for Collaborative LLMs"
_ICLR.cc/2026/Conference — ICLR 2026 Poster_

### Official Review · Reviewer_GGUj · 2025-10-28

**Soundness:** 3
**Presentation:** 3
**Contribution:** 3
**Rating:** 6
**Confidence:** 4

**Summary:**

This paper proposes AT-GRPO, which addresses the problem of applying on-policy RL to MAS with LLMs to improve their collaborative capabilities. This method achieves significant improvements on the Qwen3 model across four tasks: game playing, planning, coding, and mathematics.

**Strengths:**

Overall, this paper addresses the question of 'why standard GRPO fails in MAS' and proposes a novel sampling and grouping strategy based on the GRPO algorithm, which is applicable to MAS and provides a supporting engineering implementation.

The experiments are logically clear and demonstrate the advantages of the algorithm and the effectiveness of its components under different parameter base models and task domains.

**Weaknesses:**

In this paper, AT-GRPO uses K sampling and greedy selection of the optimal action in the rollout phase. This effectively provides a search budget of K for each decision step. In contrast, the baseline does not appear to have this budget.

I have doubts about the fairness of this comparison. How much of the performance improvement is due to AT-GRPO's RL update algorithm, and how much is due to the K-fold increase in the search budget during rollout?

Furthermore, the paper relies on GRPO's variance reduction properties, but does not theoretically analyze whether the new "tree sampling + greedy execution" strategy introduces new biases, or discuss the stability of its advantage estimate as T and K increase.

**Questions:**

1.	All current experiments are based on two agents. Has the scalability to the number of agents been examined, including performance and computational overhead?

2.	Similarly, the paper repeatedly emphasizes the significant gains on "long-horizon planning tasks" in the abstract and conclusions. However, the turn horizon in the experimental setting is set to T=4.

3.	There is a lack of strong comparison with other related work on MAS-RL, such as MAPoRL and CURE.

---

> ### Author Response · Authors · 2025-11-22
>
> We thank the reviewers for their valuable feedback and for recognizing our novel strategy, comprehensive experimental analysis, and practical engineering. The main concerns include the need for a more detailed description of our experimental design, a more detailed explanation of the experimental results, and missing baseline experiments. We uploaded a revised version according to review comments.
>
> ### **For Weakness: the doubts about K-sampling and missing ablation study.**
>
> 1. Both the SA and MAS baselines use the **same K-sampling scheme** during rollout: at each decision step we sample K candidates and execute the greedy choice among them. We have updated Sec. 5.1.1 to state this explicitly, so the performance gains of AT-GRPO are not due to AT-GRPO enjoying a larger sampling budget.
>
> 2.  **AT-GRPO vs. parallel MAS + GRPO.**
>    To draw distinction between our update rule and generic GRPO, we add a “MAS + GRPO” baseline with parallel sampling (see Tables R1–R2 and Sec. 5.2). Results show that naively grouping parallel rollouts is clearly inferior to our tree-based AT-GRPO: for example, on Qwen3-8B (Table R2), MAS + GRPO remains at 30.00 on Sokoban and 33.30 on AIME24, whereas AT-GRPO attains 98.00 and 57.00, respectively. This substantial gap confirms that comparing trajectories with divergent histories destabilizes the optimization, rendering parallel sampling ineffective for complex multi-turn MAS.
>
> **Table R1: Qwen3 1.7B results MAS + GRPO vs. MAS + AT-GRPO subset.**
>
> | Method | Sudoku | Sokoban | Plan-Path | LiveCodeBench | APPS | CodeContests | AIME24 | AIME25 | OlympiadBench |
> |---|---:|---:|---:|---:|---:|---:|---:|---:|---:|
> | MAS | 69.00 | 0.00 | 10.00 | 19.00 | 16.60 | 3.60 | 13.30 | 13.00 | 35.90 |
> |$\underline{\text{MAS + GRPO}}^*$ | 87.00 | 1.00 | 82.00 | 20.60 | 17.60 | 4.80 | 13.30 | 16.70 | 35.00 |
> | MAS + AT-GRPO w/ shared policy | **99.00** | 10.00 | 96.00 | 20.90 | 17.60 | 4.80 | **16.70** | 16.70 | **39.60** |
> | MAS + AT-GRPO w/ per-role policies | **99.00** | **11.50** | **97.00** | **24.00** | **18.60** | **7.80** | 13.30 | **18.30** | 35.20 |
>
> **Table R2: Qwen3 8B results MAS + GRPO vs. MAS + AT-GRPO subset.**
>
> | Method | Sudoku | Sokoban | Plan-Path | LiveCodeBench | APPS | CodeContests | AIME24 | AIME25 | OlympiadBench |
> |---|---:|---:|---:|---:|---:|---:|---:|---:|---:|
> | MAS | 72.00 | 16.00 | 71.00 | 28.00 | 44.40 | 17.60 | 36.60 | 30.00 | 56.50 |
> | $\underline{\text{MAS + GRPO}}^*$ | 99.00 | 30.00 | 96.00 | 24.20 | 40.20 | 10.30 | 33.30 | 26.67 | 53.20 |
> | MAS + AT-GRPO w/ shared policy | **99.50** | 96.00 | 93.00 | 30.28 | 45.80 | **18.10** | 50.00 | 35.20 | **56.80** |
> | MAS + AT-GRPO w/ per-role policies | 99.00 | **98.00** | **96.00** | **33.10** | **46.50** | **18.10** | **57.00** | **40.00** | 56.60 |
>
> $^*$ This is our newly added baseline.

---

> > ### Author Response · Authors · 2025-11-22
> >
> > ### **For Q1: All current experiments are based on two agents. Has the scalability to the number of agents been examined, including performance and computational overhead?**
> >
> > The number of agents $N$ in our experiments is primarily dictated by the specific workflow required for the task (e.g., a collaborative pair consisting of a "Coder" and a "Tester"). We focused on this minimal collaborative setting ($N=2$) to validate the effectiveness of role specialization within a manageable computational budget.
> >
> > However, our framework is designed to support scalability beyond dyadic interactions:
> > * **Structural Scalability:** The underlying tree search implementation natively supports arbitrary multi-agent topologies ($N > 2$) without requiring architectural changes to the communication protocol.
> > * **Computational Scalability:** As detailed in the newly added **Appendix G**, we theoretically verify that the computational overhead scales linearly with the number of agents. We have included a discussion on potential applications for $N>2$ (e.g., distinct Planner/Executor/Evaluator triads) in the Future Work section.
> >
> >
> >
> > ### **For Q2: The paper emphasizes gains on "long-horizon planning tasks," yet the turn horizon in the experimental setting is set to $T=4$. Is this contradictory?**
> >
> > We thank the reviewer for pointing out this potential confusion. To clarify, the turn horizon in our experiments denotes the number of interaction rounds between agents (i.e., how many times they can debate and align), rather than the length of the action trajectory. In each agent turn the agent outputs a full multi-step action sequence (e.g., an entire move sequence or path), which can span many environment steps. Thus, the underlying planning problems we study are still long-horizon in terms of environment steps.
> >
> >
> > ### **For Q3: There is a lack of strong comparison with other related work on MAS-RL, such as MAPoRL and CURE.**
> >
> > We thank the reviewer for this constructive suggestion. In the revision, we have added a new Section 5.4 together with Table 4, where we conduct quantitative comparisons against representative MAS-RL frameworks including MAPORL, MARFT, and CURE under matched backbone models and dataset splits.
> >
> > On gsm8k with Phi-3-mini-128k, our inference-only MAS achieves 84.4% accuracy, surpassing trained MAPORL (81.0%), and further improves to 88.7% with AT-GRPO, highlighting the benefit of role specialization over homogeneous debating. For math reasoning with Qwen2.5-3B, our multi-turn MAS reaches 84.4% versus MARFT’s 78.7% and climbs to 87.1% after training, indicating that iterative error correction is more effective than single-turn preference optimization. On CodeContests, our self-refining MAS raises Pass@1 from 22.8% (vanilla) and 25.9% (CURE) to 30.3% without RL and 34.2% with AT-GRPO, demonstrating that iterative test-based debugging provides consistent advantages over single-turn code+test generation.
> >
> > Table 4: Comparison with existing MARL frameworks. We report Accuracy (%) for math/logic tasks and Pass@1 (%) for code.
> >
> > (a) vs. MAPORL (Math: gsm8k, Backbone: Phi-3-mini-128k, 3.4B)
> >
> > | Method           | Config      | Acc (%) |
> > |------------------|------------|---------|
> > | Vanilla Baseline | Zero-shot  | 65.0    |
> > | MAPORL           | Trained    | 81.0    |
> > | Ours (MAS)       | Untrained  | 84.4    |
> > | Ours (MAS + RL)  | AT-GRPO    | 88.7    |
> >
> > (b) vs. MARFT (Math: gsm8k, Backbone: Qwen2.5-3B)
> >
> > | Method           | Config      | Acc (%) |
> > |------------------|------------|---------|
> > | Vanilla Baseline | Zero-shot  | 65.0    |
> > | MARFT            | Trained    | 78.7    |
> > | Ours (MAS)       | Untrained  | 84.4    |
> > | Ours (MAS + RL)  | AT-GRPO    | 87.1    |
> >
> > (c) vs. CURE (Code: CodeContests, Backbone: Qwen2.5-7B-Instruct)
> >
> > | Method           | Config      | Pass@1 (%) |
> > |------------------|------------|------------|
> > | Vanilla Baseline | Zero-shot  | 22.8       |
> > | CURE             | Trained    | 25.9       |
> > | Ours (MAS)       | Untrained  | 30.3       |
> > | Ours (MAS + RL)  | AT-GRPO    | 34.2       |

---

> ### Author Response · Authors · 2025-11-25
> **Kind Follow-up on Our Rebuttal Submission**
>
> Dear Reviewer GGUj,
>
> We hope this message finds you well.
>
> We are writing to kindly let you know that we have posted a detailed rebuttal addressing your questions and concerns. If you have a moment, we would be truly grateful if you could check our responses and let us know if any issues remain.
>
> Specifically regarding your query about the extensibility of AT-GRPO: We fully agree that validating the method in MAS settings with more than two agents is important. While AT-GRPO extends to these settings in principle, **we are currently running new experiments to empirically demonstrate this. We are working to finalize these results and will share them with you as soon as they are ready.**
>
> With the discussion deadline approaching, we remain fully dedicated to improving our manuscript. Should you feel that any further experimental validation is critical to your final assessment, please do let us know.
>
> We deeply appreciate the time and insight you have dedicated to improving our work, and we sincerely thank you for your consideration.
>
> Warm regards,
>
> The Authors

---

### Official Review · Reviewer_2wky · 2025-10-30

**Soundness:** 2
**Presentation:** 2
**Contribution:** 1
**Rating:** 2
**Confidence:** 5

**Summary:**

This paper addresses the significant challenge of applying on-policy RL to Multi-Agent Systems composed of Large LLMs. The authors identify two primary bottlenecks. First, standard group-based RL objectives, such as GRPO, rely on an assumption of identical prompts for all samples within a comparison group. This assumption is fundamentally violated in MAS, where prompts are heterogeneous, differing by agent role and interaction turn. Second, existing RL-for-LLM training frameworks are predominantly designed for single-agent, single-policy optimization. They lack the architectural support for the complex, multi-agent rollouts and concurrent on-policy parameter updates for multiple, distinct policies required by MAS. To overcome these challenges, the authors propose a two-part solution, the AT-GRPO. This method creates valid GRPO comparison groups by branching $K$ samples at each agent's specific turn in the interaction. The algorithm is characterized by (i) tree-structured sampling, (ii) agent-and-turn-wise grouping, and (iii) an agent-wise credit assignment mechanism that utilizes a mix of global (team) and local (agent-specific) rewards.

**Strengths:**

1. The paper correctly identifies a key frontier in AI research. The field is actively moving beyond single-agent LLM fine-tuning and prompt-only MAS frameworks. The challenge of applying on-policy RL—which is often crucial for stable learning in complex, non-stationary environments—to heterogeneous, multi-policy MAS is both a real and difficult problem. The paper's attempt to provide a principled solution is commendable.

2. The paper provides a nuanced and well-reasoned analysis of the trade-off between using a single, role-sharing policy versus multiple, role-specialized policies. The finding that specialized policies are superior for roles with highly distinct functions (e.g., the Coder and Tester in the code domain) is intuitive but empirically validated. Conversely, the finding that a shared policy can sometimes be superior for roles with overlapping skills (e.g., the Reasoner and Tool agent in the math domain, where the Tool agent still needs to reason) is a practical and useful takeaway for researchers and practitioners in this field.

3. The practical engineering of the MAS training system, while derivative of prior work (HybridFlow), is a non-trivial and necessary contribution to enable this line of research. More importantly, the Appendix is exemplary in its thoroughness. It provides exhaustive details on the complex, task-specific reward functions for all five domains (A.1.1-A.1.5) and the complete prompt templates for every agent, role, and interaction phase (A.2.2). This level of transparency is critical for reproducibility and is highly commendable.

**Weaknesses:**

1. The paper's headline claim—that AT-GRPO boosts planning accuracy from ~14-47% to 96-99%—appears to be built on a comparison between two fundamentally different tasks.

- Baseline Definition: The single-agent (SA) baseline for Plan/Game tasks is defined as "one agent outputs a plan (same termination condition)". For the Code baseline, it is "one agent emits code; single-turn termination". This implies a one-shot, open-loop plan generation.
- MAS Workflow Definition: The Multi-Agent System, by contrast, is an iterative feedback loop: "Planner proposes actions; Executor calls tools and returns effects/observations... termination when the goal is met or turns reach K".

This is a comparison between apples and oranges. The SA baseline is given an exponentially harder open-loop planning problem, while the MAS method is given a standard, iterative, closed-loop planning problem where it receives state feedback at every turn. The massive performance gain (e.g., 14.0% -> 96.0% on Plan-Path ) is likely dominated by this difference in task workflow, not the RL algorithm. The paper is currently attributing the gains of the MAS workflow to the AT-GRPO algorithm.

2. The paper's remarkable achievements on planning tasks (96-99% accuracy) are fatally confounded. These results are not the product of the AT-GRPO algorithm, but rather of large-scale, complex, task-specific reward functions. These reward functions (see Appendix A.1 for details) provide the agent with "oracle-level" guidance. The algorithm is merely learning how to optimize a pre-computed solution.
- Plan-Path $s_{sp,k}^{Planner}$: The reward component is based on a function called SPNEXT. A positive reward is given if and only if the agent's action $a_k$ is "on at least one shortest path from $s_{k-1}$ to the goal". This is, by definition, a shortest path oracle. The agent is not "learning to plan"; it is simply rewarded at each step for following "breadcrumbs" provided by an external oracle that has already solved the problem.
- Sokoban $s_{dlk,k}^{Planner}$: This component is based on a DEADLOCKFREE heuristic, rewarding the agent for moves that "avoid standard static corner deadlocks." The whole difficulty of Sokoban lies in identifying and avoiding such deadlocks. It provides dense, step-by-step "breadcrumbs"—which, by definition, is a shortest path oracle—for each step. The agent isn't "learning to plan"; it's simply rewarded at each step for following "breadcrumbs" provided by an external oracle that has already solved the problem. This step-by-step reward makes the long-term credit allocation problem insignificant.
- Code $s_{cov,k}^{Tester}$: Testers are rewarded based on the "mutation score" of their generated tests on "golden code." This requires not only a golden reference implementation but also a complete mutation testing framework. This is an extremely powerful and expensive validator, far exceeding simple "pass/fail" signals.

3. The paper's presentation of its reward mechanism is contradictory and misrepresents a critical, hand-tuned component of the method. Sec 5.1 explicitly states: "The reward-mixing coefficient is $\alpha=1$ without further tuning". Equation 4 1 defines $\alpha$ as the weight on the team reward ($r^{team}$). This claim implies the method uses 100% team reward and requires no task-specific reward tuning. This claim is factually incorrect. The appendix, which details the actual reward designs, reveals complex, hand-picked, task-specific mixing coefficients (which are renamed $\lambda$): $\lambda_{math} = 0.70$, $\lambda_{sudoku} = 0.60$, $\lambda_{plan} = 0.50$, $\lambda_{sok} = 0.40$.

**Questions:**

See Weaknesses

---

> ### Author Response · Authors · 2025-11-22
>
> We thank the reviewers for their valuable feedback and for recognizing our research value, comprehensive experimental analysis, and practical engineering of the MAS training system. The main concerns include the need for a more detailed description of our experimental design and a more detailed explanation of the experimental results. We uploaded a revised version according to review comments. Most importantly, we clarify below that our experimental setting is a fair and robust comparison against baselines. Our setup is summarized as follows:
> | Domain | Setting | Roles & Interaction | Turns | Environment Observation | Reward | Training Context | Inference Context |
> | :--- | :--- | :--- | :--- | :--- | :--- | :--- | :--- |
> | **Code** | **SA** | Coder (Direct Generation) | 1 (Multi-turn setting see Appendix F) | Problem Description | Ground Truth Verification | $\{Obs, Reward\}$ | $\{Obs\}$ |
> | | **MA** | Coder + Tester (Debating) | Multi | Problem Description |  Ground Truth Verification | $\{Obs, Reward\}$ | $\{Obs\}$ |
> | **Math** | **SA** | Reasoner (Direct Reasoning) | 1 (Multi-turn setting see Appendix F) | Problem Statement | Ground Truth Verification | $\{Obs, Reward\}$ | $\{Obs\}$ |
> | | **MA** | Reasoner + Tool-User (Debating) | Multi | Problem Statement | Ground Truth Verification | $\{Obs, Reward\}$ | $\{Obs\}$ |
> | **Planning** | **SA** | Executor Only | Multi | Current Game State | Goal Satisfaction | $\{Obs, Reward\}$ | $\{Obs\}$ |
> | | **MA** | Executor + Tool-User (Collab.) | Multi | Current Game State | Goal Satisfaction | $\{Obs, Reward\}$ | $\{Obs\}$ |
>
> ### **For W1: The paper's headline claim, that AT-GRPO boosts planning accuracy from ~14-47% to 96-99%, appears to be built on a comparison between two fundamentally different tasks.**
>
> We understand that our original description of the Single-Agent (SA) workflow in Section 5.1.3 might be confusing. We refine our writing in Section 5.1.3 and wish to clarify that our comparison is fair, aligned strictly on environmental feedback. We clarify the experiment settings for the two distinct task types below:
>
> 1. For **planning and gaming tasks** the only difference between the MA and SA settings in our implementation is that MA uses multiple (two in our case) agents. Both settings adopt a multi-turn workflow since planning and gaming environment provide well-defined state, action and rewards, all of which are kept identical in MA and SA settings.
> 2. For **coding and math tasks**, the MA system's multi-turn structure relies on feedback from internal agent communication, not external signals. On the other hand, SA doesn't have this signal for multi-turn interaction. Following the settings in SETS [1], we evaluated a Multi-Turn SA variant (self-refinement).
>
> To quantitatively assess whether simply granting a single agent more turns can bridge the gap to MAS, we perform controlled ablations on Qwen3 1.7B (Table R4) and 8B (Table R5) across Code and Math benchmarks under single-turn vs. multi-turn SA workflows, with and without RL. As shown below, the multi-turn SA variants (Self-Refinement style) fail to provide consistent gains, and often degrade performance, while RL mainly improves the standard single-turn SA setting.
>
> **Table R4: Single-agent ablations on Code and Math (Qwen3 1.7B).**
>
> | Setting              | LiveCodeBench | APPS | CodeContests | AIME24 | AIME25 | Olympiad |
> | -------------------- | ------------- | ---- | ------------ | ------ | ------ | -------- |
> | SA, single turn      | 11.6          | 16.2 | 3.6          | 13.4   | 9.8    | 22.2     |
> | SA + multi-turn      | 10.4          | 10.4 | 0.0          | 3.3    | 6.7    | 15.8     |
> | SA, single turn + RL | 18.8          | 17.0 | 3.0          | 10.0   | 6.7    | 23.8     |
> | SA, multi-turn + RL  | 17.7          | 13.3 | 1.2          | 6.67   | 3.3    | 16.9     |
>
> **Table R5: Single-agent ablations on Code and Math (Qwen3 8B).**
>
> | Setting              | LiveCodeBench | APPS | CodeContests | AIME24 | AIME25 | Olympiad |
> | -------------------- | ------------- | ---- | ------------ | ------ | ------ | -------- |
> | SA, single turn      | 22.8          | 30.2 | 15.75        | 18.3   | 20.0   | 55.0     |
> | SA + multi-turn      | 7.8           | 20.3 | 5.12         | 16.7   | 16.7   | 53.4     |
> | SA, single turn + RL | 25.7          | 37.0 | 12.12        | 18.3   | 26.67  | 54.8     |
> | SA, multi-turn + RL  | 16.8          | 35.4 | 11.1         | 16.7   | 23.3   | 51.2     |
>
> We provide further details and discussion in Appendix F, this variant fails to improve performance (and often degrades it) due to the lack of new signals and the base model’s limitation in making self-correction.
>
> [1] Jiefeng Chen, Jie Ren, Xinyun Chen, Chengrun Yang, Ruoxi Sun, Jinsung Yoon, and Sercan Ö. Arık. "SETS: Leveraging Self-Verification and Self-Correction for Improved Test-Time Scaling." arXiv preprint arXiv:2501.19306 (2025).

---

> ### Author Response · Authors · 2025-11-22
>
> ### **For W2: These results are not the product of the AT-GRPO algorithm, but rather of large-scale, complex, task-specific reward functions.**
> **(a) For Plan-Path and Sokoban, the agent is not "learning to plan"; it is simply rewarded at each step for following the oracle**
>
>  We respectfully clarify both points below.
> 1. *Our training does not imitate a predefined oracle.*
>    - (a) The training and test environments are different. As detailed in Sec. 5.1.4 that elaborates on the training dataset, the maze variants used for training and evaluation are disjoint sets. Thus, the agent cannot simply memorize a fixed shortest path; it must learn a policy that generalizes to unseen layouts.
>    - (b) We apply the same dense reward for Single-Agent (SA) and Multi-agent System (MAS). The SA GRPO baseline in Table 1 is trained with exactly the same dense reward functions as the MAS settings. If these rewards alone acted as a “pre-computed solution,” a single agent should also reach high accuracy. In practice, however, the SA baseline still fails to learn effectively (12.0% on Plan-Path and 48.0% on Sokoban), whereas our MAS + AT-GRPO achieves 93–96%. This gap indicates that the gains come from the MAS workflow and AT-GRPO’s credit assignment, not from the reward definition alone.
>
> 2. *AT-GRPO does not rely on dense reward shaping.* To directly test the dependence on dense heuristics, we conducted an ablation with a strict outcome-only, binary reward (see Table R3 below, also we provide further details in Appendix E). In this setting, all intermediate heuristics are removed—no shortest-path checks, no deadlock detection—leaving only a binary terminal success indicator.
>
> **Table R3: Performance Comparison with Sparse Outcome-Only Rewards**
>
> | Task      | SA baseline | MAS baseline | AT-GRPO (Outcome-only) | AT-GRPO (Dense, original) | Robustness (Drop Δ) |
> | --------- | ----------- | ------------ | ---------------------- | ------------------------- | ------------------- |
> | Sokoban   | 48.0%       | 72.0%        | **93.0%**              | 96.0%                     | -3.0%               |
> | Sudoku    | 9.0%        | 16.0%        | **99.5%**              | 99.5%                     | **0.0%**            |
> | Plan-Path | 12.0%       | 71.0%        | **89.0%**              | 93.0%                     | -4.0%               |
>
> From the results we see with only outcome-based reward, on gaming envrionment Sokoban, AT-GRPO still reaches 89.0% accuracy (vs. the SA baseline’s 12.0%, which remains low with or without dense rewards). The drop from 93% → 89% suggests that dense shaping modestly improves sample efficiency, but is not the root cause of the strong performance.
>
> Moreover, using outcome-based reward, on long-horizon planning tasks, AT-GRPO maintains near-SOTA performance (93.0% on Sokoban and 99.5% on Sudoku). This shows that our method continues to solve these tasks even when the step-by-step “breadcrumbs” are removed and long-term credit assignment becomes genuinely sparse.
>
> Taken together, these results show that:
> - the dense rewards we use are standard RL heuristics rather than oracle demonstrations.
>
> - AT-GRPO, combined with the MAS workflow, can learn effective planning policies even under purely sparse, outcome-only rewards. As summarized in Table R2 below, AT-GRPO remains highly effective across long-horizon planning tasks, with only modest drops relative to dense rewards while still substantially outperforming both SA and MAS baselines.
>
> **(b) For Code: Testers are rewarded based on the "mutation score"  which requires not only a golden reference implementation but also a complete mutation testing framework. This is an extremely powerful and expensive validator.**
>
> We thank the reviewer for raising this point. In the original draft, we described the Tester signal as a “mutation score on golden code”. This phrasing was somewhat ambiguous and could be read as assuming a full mutation-testing framework (with many mutants, mutation operators, etc.). Our actual implementation, however, is much simpler: in all experiments, we only use the pass rate of the generated unit tests when executed on a fixed reference implementation (“golden code”).
>
> In the revised version, we clarify the Tester’s non-run reward as follows. At turn \(k\), we compute the fraction of tests that pass on the golden code as the reward signal. Importantly, this is exactly the signal used in all reported experiments. The revised manuscript refines the terminology in Appendix B2 (replacing the ambiguous phrase “mutation score” with “test-pass rate on the golden code”) to avoid confusion; the underlying algorithm and experimental setup remain unchanged.

---

> ### Author Response · Authors · 2025-11-22
>
> ### **For W3: The paper's presentation of its reward mechanism is contradictory for the reward-mixing coefficient.**
>
> We clarify two misunderstandings regarding the reward structure:
>
> 1. **Interpretation of Eq. (4).**
>    The per-agent reward for turn $t$ is defined as:
>   $r_{t,i} = \alpha \cdot r^{\mathrm{team}} + r^{\mathrm{loc}}_{i}.$
>
>    Setting ($\alpha$ = 1) simply assigns unit weight to the team component; it **does not** remove the local term $(r^{loc}_{i})$. Thus, our method never degenerates to “100% team reward”.
>
> 2. **Appendix discrepancy.**
>    The task-specific coefficients $\lambda$ in the appendix were remnants of a deprecated design and were **never used** in our reported experiments. All results were obtained with a uniform configuration ($\alpha$ = 1) without any per-task tuning. In the revision, we have removed these vestigial definitions so that the appendix strictly aligns with the main text and the released code.

---

> ### Author Response · Authors · 2025-11-25
> **Kind Follow-up on Our Rebuttal Submission**
>
> Dear Reviewer 2wky,
>
> We hope this message finds you well.
>
> We are writing to kindly let you know that we have posted a detailed rebuttal addressing your questions and concerns. If you have a moment, we would be truly grateful if you could check our responses and let us know if any issues remain.
>
> With the discussion deadline approaching, we remain fully dedicated to improving our manuscript. Should you feel that any further experimental validation is critical to your final assessment, please do let us know.
>
> We deeply appreciate the time and insight you have dedicated to improving our work, and we sincerely thank you for your consideration.
>
> Warm regards,
>
> The Authors

---

> > ### Comment · Reviewer_2wky · 2025-11-28
> > **Thanks for authors' responses**
> >
> > I decided to raise my score since authors have addressed most of my concerns regarding experiments.
> >
> > Just one more question, could AT-GRPO be applied to MAS with the agent number larger than 2 (maybe 3 or more, ideally should be larger than 5 if possible for more complex tasks)? If so, have authors tried that? It is not easy to illustrate the scope of application of AT-GRPO with only two agents.

---

> > > ### Author Response · Authors · 2025-11-28
> > >
> > > Dear Reviewer 2wky,
> > >
> > > We are glad that our clarification and the additional experiment have addressed most of your concerns!
> > >
> > > Thank you for raising this question. In principle, AT-GRPO extends to other MAS settings with more than 2 agents. We are currently running these experiments and will share the results as soon as they are ready.
> > >
> > > Best,
> > >
> > > Authors

---

### Official Review · Reviewer_SCZc · 2025-10-31

**Soundness:** 2
**Presentation:** 3
**Contribution:** 2
**Rating:** 4
**Confidence:** 4

**Summary:**

This paper focus on developing on-policy RL to Multi-Agent Systems of LLMs. This work introduces AT-GRPO, which uses an "Agent- and Turn-wise" grouped tabular to store agent- and turn-wise state values. This work also developed on-policy training for shared/separated LM parameters, based on VeRL.

Experiments across four domains (game, planning, coding, math) reveal a dramatic bifurcation: near-perfect accuracy on long-horizon planning tasks (96–99.5%), but modest gains on complex reasoning (3–18% avg. improve in code/math).

**Strengths:**

1. The manuscript is well-written and easy to follow.

2. The authors have invested considerable effort in designing an effective RL training framework for multi-agent systems (MAS), adeptly handling both shared and distinct LLM parameters.

3. Comprehensive experiments were conducted across two model scales (Qwen-3 1.7B and 8B).

**Weaknesses:**

1. The experiments use an unfair reward engineering for the proposed method: AT-GRPO got 96-99.5% on planning/games while the single agent GRPO baseline only got 0-56%. A major reason is the local reward functions detailed in Appendix A.1, they provide lots of oracle information to the algorithm, e.g. in Plan-Path, the local reward is 1 if and only if the agent's chosen action "lies on at least one shortest path from $s_{k-1}$ to goal." **This is not learning to reason; this is learning to imitate an predefined oracle.** In contrast, no detailed reward function is provided for the single-agent GRPO baseline, suggesting it was unfairly handicapped and may have only used a sparse final reward.

2. The proposed Agent- and Turn-wise Group-normed advantage is a very natural extention of tabular-wise value estimate to MAS, like GiGPO[1] from GRPO, which is not mentioned in this paper. And consider the literature of cooperative MARL, tabular-wise value estimate may not be a reasonable credit assignment, a common practice in this field is to use CTDE methods like MAPPO and MADDPG which introduce a global value function for credit assignment.

3. The author should conduct ablations on the Agent-wise and Turn-wise design of the advantage, e.g. an experiment of MAS + GRPO.

4. This paper also missing some import MARL baselins for LLM, e.g. MAPoRL mentioned in the related work. Additionally this paper should also discussed some related works like MARTI[2] and MARFT[3]

[1] Feng, Lang, et al. "Group-in-group policy optimization for llm agent training." arXiv preprint arXiv:2505.10978 (2025).

[2] https://github.com/TsinghuaC3I/MARTI

[3] Liao, Junwei, et al. "Marft: Multi-agent reinforcement fine-tuning." arXiv preprint arXiv:2504.16129 (2025).

**Questions:**

1. What is the computational and memory complexity of maintaining the "Agent- and Turn-wise" advantage grouping (Algorithm 1, line 8)? And how does this approach scale as the number of agents ($N$) and the turn horizon ($T$) increase in more complex MAS environments?

2. Tables 1 & 2 show that the "MAS (prompt-only)" baseline significantly outperforms the "Single Agent (prompt-only)" baseline, even without any RL training. Sec 5.1 mentioned that for code and math, the SA baseline is "single-turn termination" while the MAS baseline is allowed $T=4$ turns. Is this performance gap primarily an artifact of this unfair comparison, where the MAS baseline benefits from multiple turn iterative refine while the SA baseline does not?

---

> ### Author Response · Authors · 2025-11-22
>
> We thank the reviewers for their valuable feedback and for recognizing our efficient training framework, comprehensive experimental design. The main concerns include the explanation of the experiment's setting and ablation studies. We uploaded a revised version according to review comments. Most importantly, we clarify below that our experimental setting is a fair and robust comparison against baselines. Our setup is summarized as follows:
> | Domain | Setting | Roles & Interaction | Turns | Environment Observation | Reward | Training Context | Inference Context |
> | :--- | :--- | :--- | :--- | :--- | :--- | :--- | :--- |
> | **Code** | **SA** | Coder (Direct Generation) | 1 (Multi-turn setting see Appendix F) | Problem Description | Ground Truth Verification | Obs, Reward | Obs|
> | | **MA** | Coder + Tester (Debating) | Multi | Problem Description |  Ground Truth Verification | Obs, Reward | Obs|
> | **Math** | **SA** | Reasoner (Direct Reasoning) | 1 (Multi-turn setting see Appendix F) | Problem Statement | Ground Truth Verification |Obs, Reward | Obs|
> | | **MA** | Reasoner + Tool-User (Debating) | Multi | Problem Statement | Ground Truth Verification | Obs, Reward | Obs|
> | **Planning** | **SA** | Executor Only | Multi | Current Game State | Goal Satisfaction | Obs, Reward | Obs|
> | | **MA** | Executor + Tool-User (Collab.) | Multi | Current Game State | Goal Satisfaction |Obs, Reward | Obs|
> ### **For W1: The experiments use an unfair reward engineering for the proposed method**
>
> We want to clarify that our training is not imitating a predefined oracle.
> - Our training and test environments are different (see Sec 5.1.4)
> - The Single-Agent (SA) GRPO baseline reported in Table 1 utilized the same dense reward functions as MA settings. Despite having access to the dense reward, the Single Agent failed to learn effectively (achieving only 12.0% on Plan-Path and 48.0% on Sokoban). We refine our experiment settings description  (see Sec 5.1.3) and highlight the same reward design in both SA and MA.
> - AT-GRPO does not rely on dense reward. We have conducted a new ablation study in this version using an outcome-based binary reward design (Table R3, also see Appendix E for more details). In this setting, we removed all intermediate heuristic signals (no shortest-path, no deadlock checks), leaving only a sparse binary success signal. (reward=1 when success else 0). With only outcome-based reward, AT-GRPO achieves 89.0% accuracy on Plan-Path (vs. SA's 12.0% with/without rewards). The minor drop from 93% suggests the dense reward aids convergence, but is not the root cause of success. For Sokoban & Sudoku, the method retains near-SOTA performance (93.0% and 99.5% respectively) using outcome-based reward.
>
> **Table R3: Performance Comparison with Sparse Outcome-Only Rewards**
>
> | Task      | SA baseline | MAS baseline | AT-GRPO (Outcome-only) | AT-GRPO (Dense, original) | Robustness (Drop Δ) |
> | --------- | ----------- | ------------ | ---------------------- | ------------------------- | ------------------- |
> | Sokoban   | 48.0%       | 72.0%        | **93.0%**              | 96.0%                     | -3.0%               |
> | Sudoku    | 9.0%        | 16.0%        | **99.5%**              | 99.5%                     | **0.0%**            |
> | Plan-Path | 12.0%       | 71.0%        | **89.0%**              | 93.0%                     | -4.0%               |
>
> ### **For W2: the proposed method is a very natural extension of tabular-wise value estimate to MAS. A common practice in this field is to use CTDE methods.**
>
> We agree that our Agent- and Turn-wise Group-normed advantage shares conceptual similarities with the turn-wise grouping in GiGPO, and we have added the discussion and citation in Section 4.1 of the revision. However, it is also important to note a key structural difference: GiGPO operates on terminal rewards for single agent tasks, necessitating a tabular-wise value estimate to propagate signals back to intermediate steps. In contrast, our framework applies to MAS and leverages environment-verified rewards at each turn (e.g., unit-test pass rates, step-wise correctness). Our mixed reward design with team-wise and local environmental signals follows the cooperative MARL literature (e.g., Mao et al., 2020), which shows that combining global and local signals yields the best empirical performance. This dense, environment-verified supervision allows us to compute an unbiased policy gradient directly, without the need for a learned value function or a centralized critic (as seen in CTDE methods like MAPPO).

---

> ### Author Response · Authors · 2025-11-22
>
> ### **For W3: The author should conduct ablations on the Agent-wise and Turn-wise design of the advantage, e.g. an experiment of MAS + GRPO.**
>
> We thank the reviewer for this valuable suggestion. We have conducted the requested ablations on the Agent-wise and Turn-wise designs, specifically adding the MAS + GRPO experiment. The results are reported in Table R1 and Table R2. We observe that naive GRPO yields suboptimal performance compared to our method. A detailed analysis of these findings is provided in the Results and Analysis section. Notably, Qwen3-8B exhibits suboptimal results in CodeContests (17.60 $\to$10.30) and OlympiadBench (56.50 $\to$ 53.20). As multi-turn interaction histories diverge, the group-averaged baseline incorrectly aggregates these heterogeneous states.
>
> **Table R1: Qwen3 1.7B results MAS + GRPO vs. MAS + AT-GRPO subset.**
>
> | Method | Sudoku | Sokoban | Plan-Path | LiveCodeBench | APPS | CodeContests | AIME24 | AIME25 | OlympiadBench |
> |---|---:|---:|---:|---:|---:|---:|---:|---:|---:|
> | MAS | 69.00 | 0.00 | 10.00 | 19.00 | 16.60 | 3.60 | 13.30 | 13.00 | 35.90 |
> | MAS + GRPO | 87.00 | 1.00 | 82.00 | 20.60 | 17.60 | 4.80 | 13.30 | 16.70 | 35.00 |
> | MAS + AT-GRPO w/ shared policy | **99.00** | 10.00 | 96.00 | 20.90 | 17.60 | 4.80 | **16.70** | 16.70 | **39.60** |
> | MAS + AT-GRPO w/ per-role policies | **99.00** | **11.50** | **97.00** | **24.00** | **18.60** | **7.80** | 13.30 | **18.30** | 35.20 |
>
> **Table R2: Qwen3 8B results MAS + GRPO vs. MAS + AT-GRPO subset.**
>
> | Method | Sudoku | Sokoban | Plan-Path | LiveCodeBench | APPS | CodeContests | AIME24 | AIME25 | OlympiadBench |
> |---|---:|---:|---:|---:|---:|---:|---:|---:|---:|
> | MAS | 72.00 | 16.00 | 71.00 | 28.00 | 44.40 | 17.60 | 36.60 | 30.00 | 56.50 |
> | MAS + GRPO | 99.00 | 30.00 | 96.00 | 24.20 | 40.20 | 10.30 | 33.30 | 26.67 | 53.20 |
> | MAS + AT-GRPO w/ shared policy | **99.50** | 96.00 | 93.00 | 30.28 | 45.80 | **18.10** | 50.00 | 35.20 | **56.80** |
> | MAS + AT-GRPO w/ per-role policies | 99.00 | **98.00** | **96.00** | **33.10** | **46.50** | **18.10** | **57.00** | **40.00** | 56.60 |
>
>
> ### **For W4: this paper also misses some important MARL baselines for LLM, e.g,. MAPoRL mentioned in the related work.**
>
> We sincerely thank the reviewer for pointing out these important MARL baselines. We agree that positioning our work against other RL for MAS methods is crucial. In the revised manuscript, we have significantly expanded our discussion to address this: **Sec 2 (Related Work) and Appendix A**: We have systematically compared the feature characteristics of our framework against these baselines. **Sec 5.4 and Table 4**: We added a detailed comparative experiment analysis discussing the performance distinctions between our proposed framework and existing MARL-LLM approaches, specifically MAPoRL, CURE, and MARFT.
>
> Crucially, we have included a direct empirical comparison in the updated Table 4. The results demonstrate that our method (MAS+AT-GRPO) consistently outperforms these baselines. Specifically, we surpass MAPoRL by 7.7% on Phi-3-mini (88.7% vs. 81.0%) and MARFT by 8.4% on Qwen-2.5-3B (87.1% vs. 78.7%) in math tasks. Furthermore, our approach exceeds the CURE baseline on code generation tasks, achieving 34.2% on CodeContests and 35.3% on LiveCodeBench. The improvement highlights the advantages of our two distinct MAS features: *heterogeneous agent roles* and *multi-turn iterative interaction*.
>
> ### **For Q1**
>
> (a) **What is the computational and memory complexity of maintaining the "Agent- and Turn-wise" advantage grouping (Algorithm 1, line 8)?**
>
> We efficiently implement this step (Algorithm 1, line 8) by directly mapping the specific trajectory tuple $(e, i, t)$ to a unique identifier (UID), a lightweight operation requiring only constant $O(1)$ time. Since this process involves merely basic integer manipulations, its resource consumption is microscopic compared to the heavy FLOPs required for the model's loss computation and backpropagation.
>
> (b) **And how does this approach scale as the number of agents ($N$) and the turn horizon ($T$) increase in more complex MAS environments?**
>
> our framework MAS interactions as a tree, allowing it to natively support arbitrary numbers of agents ($N$) and horizon lengths ($T$). Computational Scalability: Detailed analysis and overhead upper bounds are provided in Appendix G, demonstrating that the computational cost scales linearly with the number of agents $N$ in sequential settings. In parallel settings, continuous batching allows latency to scale primarily with debate depth $T$.
>
> We provide a more detailed analysis for the scalability of our approach in the newly added Appendix G.

---

> ### Author Response · Authors · 2025-11-22
>
> ### **For Q2: Sec 5.1 mentioned that for code and math, the SA baseline is "single-turn termination" while the MAS baseline is allowed  turns. Is this performance gap primarily an artifact of this unfair comparison, where the MAS baseline benefits from multiple turn iterative refine while the SA baseline does not?**
>
> We ensure fairness by providing identical environmental observations to both SA and MAS settings: in all cases, the agent(s) only see the problem statement, and no extra side information is given to MAS. The multi-turn structure in MAS arises solely from role decomposition and cross-agent communication, whereas the standard SA baseline follows the canonical single-turn QA format used in most LLM pretraining and evaluation.
>
> To test whether “giving SA more turns” could close the gap, we further introduce a multi-turn SA variant where a single agent repeatedly self-verifies and revises its own answer without any additional environmental feedback. As shown in Table R4 and Table R5, this forced multi-turn SA brings no consistent benefit and often degrades performance (e.g., Qwen3-1.7B drops from 13.4% to 3.3% on AIME24. Qwen3-8B drops from 5.1% to 15.8% on CodeContests). This is expected: without new information, the agent’s revisions are effectively unguided and the interaction pattern drifts away from its QA-style pretraining distribution, making it prone to harmful overwriting of initially correct solutions. We have clarified this design choice in Sec. 5.1.3 and empirically validated it in Appendix F, which is consistent with recent findings on self-refinement methods such as SETS [1].
>
> **Table R4: Single-agent ablations on Code and Math (Qwen3 1.7B).**
>
> | Setting              | LiveCodeBench | APPS | CodeContests | AIME24 | AIME25 | Olympiad |
> | -------------------- | ------------- | ---- | ------------ | ------ | ------ | -------- |
> | SA, single turn      | 11.6          | 16.2 | 3.6          | 13.4   | 9.8    | 22.2     |
> | SA + multi-turn      | 10.4          | 10.4 | 0.0          | 3.3    | 6.7    | 15.8     |
> | SA, single turn + RL | 18.8          | 17.0 | 3.0          | 10.0   | 6.7    | 23.8     |
> | SA, multi-turn + RL  | 17.7          | 13.3 | 1.2          | 6.67   | 3.3    | 16.9     |
>
> **Table R5: Single-agent ablations on Code and Math (Qwen3 8B).**
>
> | Setting              | LiveCodeBench | APPS | CodeContests | AIME24 | AIME25 | Olympiad |
> | -------------------- | ------------- | ---- | ------------ | ------ | ------ | -------- |
> | SA, single turn      | 22.8          | 30.2 | 15.75        | 18.3   | 20.0   | 55.0     |
> | SA + multi-turn      | 7.8           | 20.3 | 5.12         | 16.7   | 16.7   | 53.4     |
> | SA, single turn + RL | 25.7          | 37.0 | 12.12        | 18.3   | 26.67  | 54.8     |
> | SA, multi-turn + RL  | 16.8          | 35.4 | 11.1         | 16.7   | 23.3   | 51.2     |
>
>
> Reference:
>
> [1] Jiefeng Chen, Jie Ren, Xinyun Chen, Chengrun Yang, Ruoxi Sun, Jinsung Yoon, and Sercan Ö. Arık. "SETS: Leveraging Self-Verification and Self-Correction for Improved Test-Time Scaling." arXiv preprint arXiv:2501.19306 (2025).

---

> > ### Comment · Reviewer_SCZc · 2025-11-27
> > **Thank you for detailed responses**
> >
> > I appreciate the detailed rebuttal and additional experiments. My conern has been addressed, and I will consider adjusting my score.

---

> > > ### Author Response · Authors · 2025-11-28
> > > **Thank you for considering raising the score**
> > >
> > > Thank you for your positive feedback and for considering adjusting the score; we are glad to hear that our rebuttal addressed your concerns.

---

> ### Author Response · Authors · 2025-11-25
> **Kind Follow-up on Our Rebuttal Submission**
>
> Dear Reviewer SCZc,
>
> We hope this message finds you well.
>
> We are writing to kindly let you know that we have posted a detailed rebuttal addressing your questions and concerns. If you have a moment, we would be truly grateful if you could check our responses and let us know if any issues remain.
>
> With the discussion deadline approaching, we remain fully dedicated to improving our manuscript. Should you feel that any further experimental validation is critical to your final assessment, please do let us know.
>
> We deeply appreciate the time and insight you have dedicated to improving our work, and we sincerely thank you for your consideration.
>
> Warm regards,
>
> The Authors

---

### Official Review · Reviewer_aB72 · 2025-11-01

**Soundness:** 2
**Presentation:** 2
**Contribution:** 2
**Rating:** 4
**Confidence:** 4

**Summary:**

This paper introduces AT-GRPO, an agent- and turn-wise grouped reinforcement learning method, alongside a novel training system, to enable effective on-policy RL for LLM-based multi-agent systems. Through extensive experiments across gaming, planning, coding, and math tasks, the authors demonstrate that their approach consistently enhances performance, particularly in long-horizon planning where it elevates success rates from 14-47% to over 96%.

**Strengths:**

- The experimental design is comprehensive, and the results consistently demonstrate the effectiveness of the method. However, the model scales used (1.7B, 8B) are relatively small, leaving uncertainty about how well the approach would scale to much larger models.

- The finding that the choice between a role-sharing policy and role-specialized policies depends on the task characteristics is a very interesting and valuable insight for the field.

**Weaknesses:**

- The paper is at times hard to read. Key explanations, such as the components of Figure 3, are unclear.

- See Questions.

**Questions:**

- The description of Figure 3 is unclear. Where is the “middle” section referring to the code debugging task (mentioned around line 214)? Also, what do the gray and blue colors represent in the figure?

- The notation used across sections is inconsistent. For example, Equation 1 uses index  l \in \{1, \dots, K\}, while Algorithm 1 uses c \in \{1, \dots, K\}. This inconsistency hinders readability.

- How is the number of agents \( N \) determined in the MAS setup?

- What is the advantage of tree sampling over parallel sampling?  Additionally, in Algorithm 1, greedily selecting the action with the maximum turn-level reward at each turn t may lead to suboptimal final outcomes. Could this myopic selection affect the overall reward?

- Equation 3 does not include a clipping mechanism to constrain policy distribution changes. Is there a specific reason for omitting this common GRPO stabilization technique?

---

> ### Author Response · Authors · 2025-11-22
>
> We thank the reviewers for their valuable feedback and for recognizing our methodological novelty, comprehensive experimental design, and interesting findings. The main concerns include unclear figures, explanations, notation, and missing baseline experiments. Our responses are provided below:
>
> ### **For Weakness, Q1 and Q2**
>
> We realize that the original version of Fig. 3  lacks clarity. In the revision, we have improved Fig. 3 to clarify both the caption and the main components: gray squares now explicitly denote states and blue squares denote agent actions. We also clarified in Sec 4.1 that the “middle” section referring to the code debugging task corresponds to the multi-agent code workflows in Fig.3 (a), where a coder–tester loop iteratively refines code and unit tests until alignment (see Fig.2, top). And for notation, we have unified the notation in this version.
>
> ### **For Q3: How is the number of agents (N) determined in the MAS setup?**
>
>  In our framework, the number of agents N is specified by the task-specific workflow design (e.g., how many roles are needed, such as “Coder” and “Tester” for code debugging). Technically, the tree implementation can support arbitrary multi-agent workflows. In this work, we focus on the minimal non-trivial collaborative setting with N=2 agents, which captures meaningful role specialization while keeping the compute budget. We also discuss how the complexity scales in Appendix G.
>
> ### **For Q4**
>
> **(a) What is the advantage of tree sampling over parallel sampling?**
>
> To evaluate the advantage of tree sampling over parallel sampling, we add a "MAS + GRPO" baseline using parallel sampling（See Table R1, Table R2 and more details in Sec 5.2). Results indicate that naively grouping parallel rollouts yields significantly inferior performance. For instance, on Qwen3-8B (Table R2), MAS + GRPO stagnates at 30.00 on Sokoban and 33.30 on AIME24, whereas our tree-based method (AT-GRPO) achieves 98.00 and 57.00, respectively. This substantial gap confirms that comparing trajectories with divergent histories destabilizes the optimization, rendering parallel sampling ineffective for complex multi-turn MAS.
>
> **Table R1: Qwen3 1.7B results MAS + GRPO vs. MAS + AT-GRPO subset.**
>
> | Method | Sudoku | Sokoban | Plan-Path | LiveCodeBench | APPS | CodeContests | AIME24 | AIME25 | OlympiadBench |
> |---|---:|---:|---:|---:|---:|---:|---:|---:|---:|
> | MAS | 69.00 | 0.00 | 10.00 | 19.00 | 16.60 | 3.60 | 13.30 | 13.00 | 35.90 |
> |$\underline{\text{MAS + GRPO}}^*$ | 87.00 | 1.00 | 82.00 | 20.60 | 17.60 | 4.80 | 13.30 | 16.70 | 35.00 |
> | MAS + AT-GRPO w/ shared policy | **99.00** | 10.00 | 96.00 | 20.90 | 17.60 | 4.80 | **16.70** | 16.70 | **39.60** |
> | MAS + AT-GRPO w/ per-role policies | **99.00** | **11.50** | **97.00** | **24.00** | **18.60** | **7.80** | 13.30 | **18.30** | 35.20 |
>
> **Table R2: Qwen3 8B results MAS + GRPO vs. MAS + AT-GRPO subset.**
>
> | Method | Sudoku | Sokoban | Plan-Path | LiveCodeBench | APPS | CodeContests | AIME24 | AIME25 | OlympiadBench |
> |---|---:|---:|---:|---:|---:|---:|---:|---:|---:|
> | MAS | 72.00 | 16.00 | 71.00 | 28.00 | 44.40 | 17.60 | 36.60 | 30.00 | 56.50 |
> | $\underline{\text{MAS + GRPO}}^*$ | 99.00 | 30.00 | 96.00 | 24.20 | 40.20 | 10.30 | 33.30 | 26.67 | 53.20 |
> | MAS + AT-GRPO w/ shared policy | **99.50** | 96.00 | 93.00 | 30.28 | 45.80 | **18.10** | 50.00 | 35.20 | **56.80** |
> | MAS + AT-GRPO w/ per-role policies | 99.00 | **98.00** | **96.00** | **33.10** | **46.50** | **18.10** | **57.00** | **40.00** | 56.60 |
>
> $^*$ This is our newly added baseline.
>
> **(b) Additionally, in Algorithm 1, greedily selecting the action with the maximum turn-level reward at each turn t may lead to suboptimal final outcomes. Could this myopic selection affect the overall reward?**
>
> We thank the reviewer for this valuable feedback. Actually, the concern regarding myopic greedy selection does not apply to our framework. While greedy selection is often myopic in general RL, particularly when high immediate rewards mask low long-term value, we argue that this concern is absent in our specific setting. Our framework utilizes outcome-based verified rewards (e.g., unit-test pass rates on complete code) that are monotonically aligned with final success. Under such conditions, the immediate verified reward is consistent with the true value function $Q^\*(s, a)$. Following the principle of Bellman Optimality, since our feedback serves as a reliable proxy for the true action-value function $Q^*(s, a)$, selecting the highest-reward candidate is mathematically justified and approximates an optimal policy. Thus, our selection strategy is theoretically grounded and robust against the "high reward, low value" trap. More detailed analysis and proof can be found in Appendix B.7.

---

> > ### Author Response · Authors · 2025-11-22
> >
> > ### **For Q5: Equation 3 does not include a clipping mechanism to constrain policy distribution changes.**
> >
> > We thank the reviewer for this valuable feedback. We would like to clarify that in our actual experiments, we indeed implemented the clipping mechanism (with a clipping coefficient of $\epsilon = 0.2$) to ensure training stability. The decision to omit the clipping term in the original Equation 3 for notational brevity. However, we acknowledge that this simplification might lead to confusion. In the revised manuscript, we have updated Equation 3 to the complete formula, including the clipping mechanism.

---

> ### Author Response · Authors · 2025-11-25
> **Kind Follow-up on Our Rebuttal Submission**
>
> Dear Reviewer aB72,
>
> We hope this message finds you well.
>
> We are writing to kindly let you know that we have posted a detailed rebuttal addressing your questions and concerns. If you have a moment, we would be truly grateful if you could check our responses and let us know if any issues remain.
>
> Specifically regarding your query about the extensibility of AT-GRPO: We fully agree that validating the method in MAS settings with more than two agents is important. While AT-GRPO extends to these settings in principle, **we are currently running new experiments to empirically demonstrate this. We are working to finalize these results and will share them with you as soon as they are ready.**
>
> With the discussion deadline approaching, we remain fully dedicated to improving our manuscript. Should you feel that any further experimental validation is critical to your final assessment, please do let us know.
>
> We deeply appreciate the time and insight you have dedicated to improving our work, and we sincerely thank you for your consideration.
>
> Warm regards,
>
> The Authors

---

### Author Response · Authors · 2025-11-24

We sincerely thank all reviewers (**aB72**, **SCZc**, **2wky**, **GGUj**) for their valuable feedback and for recognizing our efficient training framework, novel method, and comprehensive experimental design. We appreciate the constructive criticism and insightful questions regarding the details of our experimental design, the explanation of results, and the need for additional baselines.

In this summary, we highlight two key updates: a clarification of our experimental settings and the introduction of a new baseline comparison.

---

### **1. Clarification of Experimental Settings**

Most importantly, we clarify below that our experimental setting provides a fair and robust comparison against baselines. To ensure fairness, we align all environmental observations and reward signals across both Multi-Agent (MA) and Single-Agent (SA) settings.

While both paradigms utilize the same role-specific reward functions, the sole distinction is that the MA framework involves multiple agents capable of discussion. Our setup is summarized as follows:

| Domain | Setting | Roles & Interaction | Turns | Environment Observation | Reward | Training Context | Inference Context |
| :--- | :--- | :--- | :--- | :--- | :--- | :--- | :--- |
| **Code** | **SA** | Coder (Direct Generation) | 1* | Problem Description | Ground Truth Verification | Obs, Reward | Obs |
| | **MA** | Coder + Tester (Debating) | Multi | Problem Description | Ground Truth Verification | Obs, Reward | Obs |
| **Math** | **SA** | Reasoner (Direct Reasoning) | 1* | Problem Statement | Ground Truth Verification | Obs, Reward | Obs |
| | **MA** | Reasoner + Tool-User (Debating) | Multi | Problem Statement | Ground Truth Verification | Obs, Reward | Obs |
| **Planning** | **SA** | Executor Only | Multi | Current Game State | Goal Satisfaction | Obs, Reward | Obs |
| | **MA** | Executor + Tool-User (Collab.) | Multi | Current Game State | Goal Satisfaction | Obs, Reward | Obs |

> \* *For details on Multi-turn settings for SA in Code and Math, please refer to Appendix F.*

---

### **2. New Baseline: MAS + GRPO**

Reviewers (**SCZc**, **GGUj**)  pointed out the need to compare our method against a standard GRPO applied to MAS to validate our specific "**Agent- and Turn-wise**" design. We have added this critical baseline (**MAS + GRPO**) using parallel sampling.

The results (detailed in **Table R1** and **Table R2**) demonstrate a substantial performance gap, proving that naive grouping fails in multi-turn multi-agent settings. As multi-turn interaction histories diverge, the group-averaged baseline incorrectly aggregates these heterogeneous states.

Notably, **Qwen3-8B** (Table R2) exhibits suboptimal results in CodeContests ($17.60 \to 10.30$) and OlympiadBench ($56.50 \to 53.20$) when using naive GRPO compared to the base MAS.

**Table R1: Qwen3 1.7B results MAS + GRPO vs. MAS + AT-GRPO subset.**

| Method | Sudoku | Sokoban | Plan-Path | LiveCodeBench | APPS | CodeContests | AIME24 | AIME25 | OlympiadBench |
|---|---:|---:|---:|---:|---:|---:|---:|---:|---:|
| MAS | 69.00 | 0.00 | 10.00 | 19.00 | 16.60 | 3.60 | 13.30 | 13.00 | 35.90 |
|$\underline{\text{MAS + GRPO}}^*$ | 87.00 | 1.00 | 82.00 | 20.60 | 17.60 | 4.80 | 13.30 | 16.70 | 35.00 |
| MAS + AT-GRPO w/ shared policy | **99.00** | 10.00 | 96.00 | 20.90 | 17.60 | 4.80 | **16.70** | 16.70 | **39.60** |
| MAS + AT-GRPO w/ per-role policies | **99.00** | **11.50** | **97.00** | **24.00** | **18.60** | **7.80** | 13.30 | **18.30** | 35.20 |

**Table R2: Qwen3 8B results MAS + GRPO vs. MAS + AT-GRPO subset.**

| Method | Sudoku | Sokoban | Plan-Path | LiveCodeBench | APPS | CodeContests | AIME24 | AIME25 | OlympiadBench |
|---|---:|---:|---:|---:|---:|---:|---:|---:|---:|
| MAS | 72.00 | 16.00 | 71.00 | 28.00 | 44.40 | 17.60 | 36.60 | 30.00 | 56.50 |
| $\underline{\text{MAS + GRPO}}^*$ | 99.00 | 30.00 | 96.00 | 24.20 | 40.20 | 10.30 | 33.30 | 26.67 | 53.20 |
| MAS + AT-GRPO w/ shared policy | **99.50** | 96.00 | 93.00 | 30.28 | 45.80 | **18.10** | 50.00 | 35.20 | **56.80** |
| MAS + AT-GRPO w/ per-role policies | 99.00 | **98.00** | **96.00** | **33.10** | **46.50** | **18.10** | **57.00** | **40.00** | 56.60 |

$^*$ This is our newly added baseline.

---

### Author Response · Authors · 2025-12-02
**Note to Area Chair**

Dear Area Chair,

Given the updated rebuttal policy for ICLR 2026, we understand that further discussion with reviewers would not be possible and scores will no longer be updated. To assist with your final assessment, we would like to briefly summarize our submission's acknowledged strengths, how we addressed the main concerns in the revised manuscript, and the status of discussions prior to the system revert.

**1. Acknowledged Strengths**
Reviewers consistently recognized the value of AT-GRPO algorithm and our implementation to enable efficient MAS training. They also commended the comprehensive experimental design and the significance of our experimental results.

**2. Addressing Key Concerns**
We have updated the manuscript to address the primary feedback:

* **Experimental Setup Clarity.** We rewrote the description of our experimental setup to clarify our baseline settings. This resolves misunderstandings regarding AT-GRPO's training efficiency and ensures the fairness of our baseline comparisons.
* **New Baselines.** To address concerns from Reviewers SCZc, aB72, and GGUj, we added a rigorous “MAS + GRPO” baseline in Section 5.2. This involves training a multi-agent system with standard GRPO but without our specific structured agent- and turn-wise advantages. We also added comparisons with other MARL frameworks for LLMs in Section 5.3.
* **Additional Experiments.** We included new experiments on agent scalability (Section 5.2) and outcome rewards (Appendix B.6) to provide a more robust evaluation.

**3. Status of Reviewer Discussions (Pre-Revert)**
Prior to the technical interruption, we engaged in constructive discussions where several concerns were effectively resolved:

* **Score Updates.** Reviewers SCZc (original: 4) and 2wky (original: 2) explicitly expressed satisfaction with our clarifications and the new MAS+GRPO results, indicating a decision to raise their scores.
* **Completion of requested experiments and clarifications.** Unfortunately, the OpenReview incident halted the discussion phase before we could further interact with reviewers aB72 and GGUj. Nevertheless, we have completed all experiments requested by the reviewers and incorporated the corresponding results and writing clarifications into the revised manuscript, fully addressing the specific concerns raised by aB72 and GGUj.

---

### Meta-Review · Area_Chair_9SdG · 2025-12-16

**Summary:**

This paper introduces AT-GRPO, an agent- and turn-wise grouped on-policy RL algorithm and a supporting MAS training system for LLM agents. Reviewers raised substantial concerns about (i) fairness of comparisons against single-agent baselines (especially planning/game workflows), (ii) heavy reward engineering that might act as an oracle, (iii) lack of ablations against standard GRPO in MAS, (iv) unclear notation/figures, (v) the omission of clipping in the loss, and (vi) scalability beyond two agents and complexity. The rebuttal substantially strengthened the paper: it added a critical MAS+GRPO baseline (Tables R1–R2 and Tables 1–2), clarified that SA and MAS share the same K-sampling and dense reward designs, provided multi-turn SA ablations showing that extra turns alone do not close the gap (Appendix F; Tables R4–R5), and introduced outcome-only reward ablations demonstrating that AT-GRPO retains high performance without dense shaping (Appendix E/Table R3). It also fixed the reward-mixing inconsistency (alpha vs. deprecated lambdas), documented the clipping term, improved figures/notation, and discussed scalability and complexity (Fig. 5; Appendix G). Remaining limitations are primarily theoretical (no formal bias/variance analysis of tree sampling and greedy turn-level execution, reliance on a strong monotonicity assumption), plus modest empirical validation beyond N=2 agents. Overall, the rebuttal overcame the main objections about fairness and baselines; the methodology and engineering are solid, and the empirical gains—especially in planning—are convincing under both dense and sparse rewards. I recommend acceptance as a poster, with recognition that theoretical rigor and broader scaling evidence would be needed for higher-tier consideration. But as the initial scores are negative, I wouldn't mind if the paper is rejected.

**Reviewer Concerns:**

#### Reviewer GGUj
1. **Concern**: Potential unfairness due to AT-GRPO using K sampling and greedy selection during rollout vs baselines.
   - **Why Unresolved**: *(Resolved)*
   - **Impact on Decision**: Resolved by clarifying identical K-sampling across SA/MAS, adding MAS+GRPO ablation (Tables R1–R2) showing tree-based grouping is essential.

2. **Concern**: Bias/variance and stability implications of tree sampling + greedy execution as T and K grow.
   - **Why Unresolved**: The rebuttal provides an assumption-based justification (Appendix B.7) but no formal analysis of estimator bias or variance and no sensitivity study across larger T/K.
   - **Impact on Decision**: Moderate. Limits theoretical strength but does not invalidate empirical results.

3. **Concern**: Scalability in agent count (N>2), both performance and computational overhead.
   - **Why Unresolved**: Appendix G analyzes linear scaling and Fig. 5 presents a 7-agent experiment, but broader empirical scaling and resource implications remain lightly evaluated.
   - **Impact on Decision**: Minor. The initial evidence is positive; more comprehensive scaling results would strengthen impact.

4. **Concern**: Long-horizon claims vs turn horizon T=4.
   - **Why Unresolved**: *(Resolved)*
   - **Impact on Decision**: Resolved by clarifying that each turn emits multi-step actions; long horizon pertains to environment steps.

5. **Concern**: Missing comparisons to MAPoRL/CURE and related MAS-RL work.
   - **Why Unresolved**: *(Resolved)*
   - **Impact on Decision**: Resolved via new Section 5.3/Table 3 comparisons under matched backbones/splits.

---

#### Reviewer 2wky
1. **Concern**: Fairness of SA vs MAS workflows in planning (open-loop SA vs closed-loop MAS leading to inflated gains).
   - **Why Unresolved**: *(Resolved)*
   - **Impact on Decision**: Resolved. Authors clarified both SA/MAS are multi-turn in planning, and added multi-turn SA ablations showing extra turns alone do not help.

2. **Concern**: Heavy reward engineering may act as an oracle (shortest-path, deadlock heuristics, mutation testing).
   - **Why Unresolved**: Outcome-only ablation shows strong performance persists; mutation testing clarified to simple pass-rate on golden code. Still, some domains retain substantial shaping, and broader tasks may not offer such verifiable signals.
   - **Impact on Decision**: Moderate. Empirically mitigated, but generalization to settings without verifiable signals remains an open question.

3. **Concern**: Contradiction between alpha=1 (team reward) and per-task lambdas in the appendix.
   - **Why Unresolved**: *(Resolved)*
   - **Impact on Decision**: Resolved; deprecated lambdas removed, alpha=1 kept, clipping documented.

4. **Concern**: Scalability beyond two agents.
   - **Why Unresolved**: Some evidence added (Fig. 5), but limited breadth of domains and agent topologies.
   - **Impact on Decision**: Minor. Not a blocker given initial positive scaling.

---

#### Reviewer SCZc
1. **Concern**: Unfair reward engineering favoring MAS and unclear SA reward design.
   - **Why Unresolved**: *(Resolved)*
   - **Impact on Decision**: Resolved. SA uses same dense rewards; outcome-only ablation confirms gains are not due to oracle-level shaping.

2. **Concern**: AT-GRPO is a natural extension of tabular group-normalized advantages (GiGPO), and CTDE/centralized critic may be preferable for credit assignment.
   - **Why Unresolved**: Discussion and citations were added; however, no direct empirical comparison against CTDE baselines (e.g., MAPPO) in these domains.
   - **Impact on Decision**: Minor. The design aligns with environment-verified rewards; broader CTDE comparisons would strengthen the case.

3. **Concern**: Missing ablations (MAS+GRPO) and MARL baselines.
   - **Why Unresolved**: *(Resolved)*
   - **Impact on Decision**: Resolved via Tables R1–R2 and Section 5.3/Table 3.

4. **Concern**: Complexity and scaling in N and T.
   - **Why Unresolved**: *(Resolved)*
   - **Impact on Decision**: Resolved by Appendix G; linear scaling analysis and system design clarified.

---

#### Reviewer aB72
1. **Concern**: Unclear figures (Figure 3 components and colors) and inconsistent notation.
   - **Why Unresolved**: *(Resolved)*
   - **Impact on Decision**: Resolved. Figures/notation improved in the revision.

2. **Concern**: Advantages of tree sampling over parallel sampling; potential myopic behavior from greedy per-turn selection.
   - **Why Unresolved**: Empirically addressed via MAS+GRPO baseline showing parallel sampling harms optimization; theoretical justification relies on a strong monotonicity assumption rather than a formal proof.
   - **Impact on Decision**: Minor to moderate. Empirical evidence is strong; theoretical rigor remains limited.

3. **Concern**: Clipping missing in the loss (Eq. 3).
   - **Why Unresolved**: *(Resolved)*
   - **Impact on Decision**: Resolved. Clipping reinstated and documented.

4. **Concern**: How N is determined; scalability beyond two agents.
   - **Why Unresolved**: Task-driven N clarified; Fig. 5 shows scaling; broader validation still limited.
   - **Impact on Decision**: Minor.

**Reviewer Scores:**

#### Reviewer GGUj
- **Original Score**: 6
- **Expected Score After Discussion**: 6 or 8
- **Rationale**: Fairness/K-sampling clarified; MAS+GRPO ablations show the necessity of agent/turn-wise grouping; outcome-only rewards mitigate oracle concerns; additional baselines (MAPoRL/CURE/MARFT) and complexity analysis added. Remaining theoretical analysis gaps reduce ceiling but warrant a modest bump. 7 is a suitable score, but there is no 7.

---

#### Reviewer 2wky
- **Original Score**: 2
- **Expected Score After Discussion**: 6
- **Rationale**: Authors addressed most concerns: SA/MAS workflow fairness with multi-turn SA ablations, outcome-only reward ablation, reward-mixing inconsistency fixed, clipping documented, and initial scalability evidence provided. The reviewer explicitly stated raising their score post-rebuttal; remaining issues are limited.

---

#### Reviewer SCZc
- **Original Score**: 4
- **Expected Score After Discussion**: 6
- **Rationale**: Reward fairness clarified (same dense rewards for SA/MAS) and outcome-only ablation; MAS+GRPO ablations and MARL baseline comparisons added; complexity/scaling discussed. The reviewer noted concerns were addressed and considered adjusting the score.

---

#### Reviewer aB72
- **Original Score**: 4
- **Expected Score After Discussion**: 6
- **Rationale**: Presentation/notation improved; clipping documented; MAS+GRPO demonstrates tree sampling benefits; greedy selection justified empirically and via assumption-based argument; scalability partially evidenced.

---

### Decision · Program_Chairs · 2026-01-26

Accept (Poster)